# Learning emergent partial differential equations in a learned emergent space

Felix P. Kemeth[1], Tom Bertalan[1], Thomas Thiem[2], Felix Dietrich[3], Sung Joon Moon[2], Carlo R. Laing[4] & Ioannis G. Kevrekidis[1 ✉]

We propose an approach to learn effective evolution equations for large systems of interacting agents. This is demonstrated on two examples, a well-studied system of coupled normal form oscillators and a biologically motivated example of coupled Hodgkin-Huxley-like neurons. For such types of systems there is no obvious space coordinate in which to learn effective evolution laws in the form of partial differential equations. In our approach, we accomplish this by learning embedding coordinates from the time series data of the system using manifold learning as a first step. In these emergent coordinates, we then show how one can learn effective partial differential equations, using neural networks, that do not only reproduce the dynamics of the oscillator ensemble, but also capture the collective bifurcations when system parameters vary. The proposed approach thus integrates the automatic, data-driven extraction of emergent space coordinates parametrizing the agent dynamics, with machine-learning assisted identification of an emergent PDE description of the dynamics in this parametrization.

[1] Department of Chemical and Biomolecular Engineering, Whiting School of Engineering, Johns Hopkins University, 3400 North Charles Street, Baltimore, MD 21218, USA. [2] The Department of Chemical and Biological Engineering, Princeton University, Princeton, NJ 08544, USA. [3] Department of Informatics, School of Computation, Information, and Technology, Technical University of Munich, Boltzmannstr. 3, 85748 Garching, Germany. [4] School of Natural and Computational Sciences, Massey University (Albany), Private Bag 102-904 Auckland, New Zealand. ✉email: yannisk@jhu.edu

Modeling the dynamic behavior of large systems of interacting agents remains a challenging problem in complex systems analysis. Due to the large state space dimension of such systems, it has historically been an ongoing research goal to construct useful reduced-order models with which to collectively describe the coarse-grained dynamics of agent ensembles. Such coarse-grained, collective descriptions arise in many contexts, e.g., in thermodynamics, where interacting particles may effectively be described at the macroscopic level by temperature, pressure and density; or in kinetic theory, where collisions in the Boltzmann equation can lead to continuum descriptions, such as the Navier-Stokes equations - but also in contexts such as chemotaxis or granular flows. One important issue in this coarse-graining is to find coarse-grained observables (density fields, momentum fields, concentration fields, void fraction fields) that describe the evolution of the collective behavior in physical space. Macroscopic, effective models are then often approximated as partial differential equations (PDEs) for these fields: their time derivatives are expressed locally in terms of the local spatial derivatives of the field(s) at each point. The closures required to derive predictive models can be obtained either mathematically (with appropriate assumptions) and/or semi-empirically through experimental or computational observations.

When the interacting agents are coupled oscillator systems, their observed low-dimensional dynamics can sometimes be described as a lumped system of a few ordinary differential equations (ODEs) in terms of so-called order parameters[1–3]. For large heterogeneous systems of interacting oscillators we observe, at any given moment, a distribution of oscillator states; being able to usefully describe this evolution by a few ODEs for appropriate order parameters corresponds, conceptually, to describing the distribution evolution through a finite, closed set of a few moment equations for the distribution. The few good order parameters are here provided by the few leading moments in terms of which a closed set of model ODEs (or even stochastic differential equations) can be written. And while in some cases such a reduced description can be quite successful, there are other cases where a few ODEs will not suffice, and where one needs to write evolution equations (e.g., PDEs) for evolving field(s) of instantaneous oscillator behavior(s).

The question then naturally arises: What is a good way of parametrizing the spatial support of this evolving distribution of behaviors? Which (and how many) are the few independent, spatial variables, in the space of which we will attempt to derive evolutionary PDE models for the collective behavior evolution? In other words, when the problem does not evolve in physical space (e.g., when the oscillators are nodes in an interacting network) does there exist a useful continuum embedding space in which we can observe the behavior evolving as a spatiotemporal field? And if so, how can we detect this emergent space and its parametrizing independent coordinates in a data-driven way, based on observations of the collection of individual coupled agent dynamics? Our task thus has two components, both accomplished here in a data-driven fashion: (a) find emergent spatial coordinates in which the oscillator behavior can be (embedded and) observed as smooth spatiotemporal field evolution; and (b) once these emergent coordinates have been obtained, learn a model of the evolving dynamics, if possible in the form of a partial differential equation governing this field; that is, approximate the (pointwise) time derivative(s) of the field(s) in terms of a few local spatial derivatives of the field in the emergent independent variables.

The data-driven approximation of such evolution operators for spatiotemporal dynamics using machine learning tools (neural networks, Gaussian processes, manifold learning...) is a long-standing research endeavor - we, among others, have worked on neural network-based identification of nonlinear distributed

systems[4–6]; the subject is currently exploding in the machine learning literature, e.g.,[7,8]. The twist in our work here is that the space in which the evolution operator (that is, the PDE) will be learned (the independent variables in which the spatial derivatives will be estimated) is not known a priori but will be rather identified, in a first step, through data mining/manifold learning[9,10]. If/when such an approach is successful, it can lead to a dramatic reduction of the computational cost of simulation/prediction of the collective, coarse-grained dynamics (compared to the individual evolution of every oscillator/agent in the ensemble). This is the case when the agent ensemble is large but the set of agents can be parametrized with only a few emergent parameters. This reduced description also enables tasks (effective stability and bifurcation analysis, even control and optimization) that would be difficult or impossible to perform with the fine-scale model. More importantly, if successful and generalizable enough, this alternative description in terms of field PDEs in emergent variables, assisted by computationally mapping back-and-forth between fine and coarse descriptions, may guide a new, coarse-grained interpretation and even understanding of the system dynamics.

There may appear to be a contradiction between having fine-scale dynamics we know to involve long-range interactions (here, all-to-all coupling), and learning a model based on local interactions (here, coupling with oscillators that have nearby behavior, through local behavior derivatives in our emergent space). We will return to this issue repeatedly in the discussion below, but we mention that the learned operators are not themselves the true physics; they are but a particular, parsimonious parametrization of the long-term dynamics (after initial transients) on a much lower-dimensional slow manifold on which the collective behavior evolves. It is the low dimensionality of this manifold, and the power of embedding theorems like those of Whitney[11] and Takens[12] that enable data-driven parameterizations (as opposed to physically meaningful mechanistic interpretations) of the long-term dynamics. The many coupled local grid points underpinning a finite-difference discretization of a PDE will here play the role of the many generic observers parametrizing the relatively low-dimensional manifold on which the coarse-grained long-term dynamics and the attractors of the system are expected to live.

This approach is fundamentally different from recent approaches where the dynamics are learned in a latent space of dependent variables, typically as systems of ODEs (but also PDEs with known independent variables). Examples of these dependent variable latent spaces include learning the dynamics of spatial principal component coefficients on an inertial manifold[13] or learning an ODE in a latent space of an autoencoder using dictionaries and sparsity promoting regularization[14]. Since early works (e.g. see[15] on the Mackey-Glass equation, also Refs.[5,6,16]), learning dynamical systems from data has regained increased attention in recent years. Popular examples include (in a vast literature) sparse identification of nonlinear dynamical systems using dictionaries[17], DeepXDE[18], neural ODEs[19], LSTM neural networks[20] and PDE-net[21]. As in the latter, the emergent PDE will be learned here from discrete time data using an explicit forward Euler time integration step (in effect, training a ResNet); many other approaches are also possible (for a ResNet-like Runge-Kutta recurrent network, see Ref.[6]).

To find coordinates in which to learn the PDE description, we follow the recent work[9,22] and use diffusion maps[23,24], a nonlinear manifold learning technique. As our agent-based example, we first illustrate our approach on coupled Stuart-Landau oscillators,

$$\frac{\mathrm{d}}{\mathrm{d}t}W_k = \left(1 + i\omega_k\right)W_k - \left|W_k\right|^2 W_k + \frac{K}{N}\sum_{j=1}^{N}\left(W_j - W_k\right); \quad (1)$$

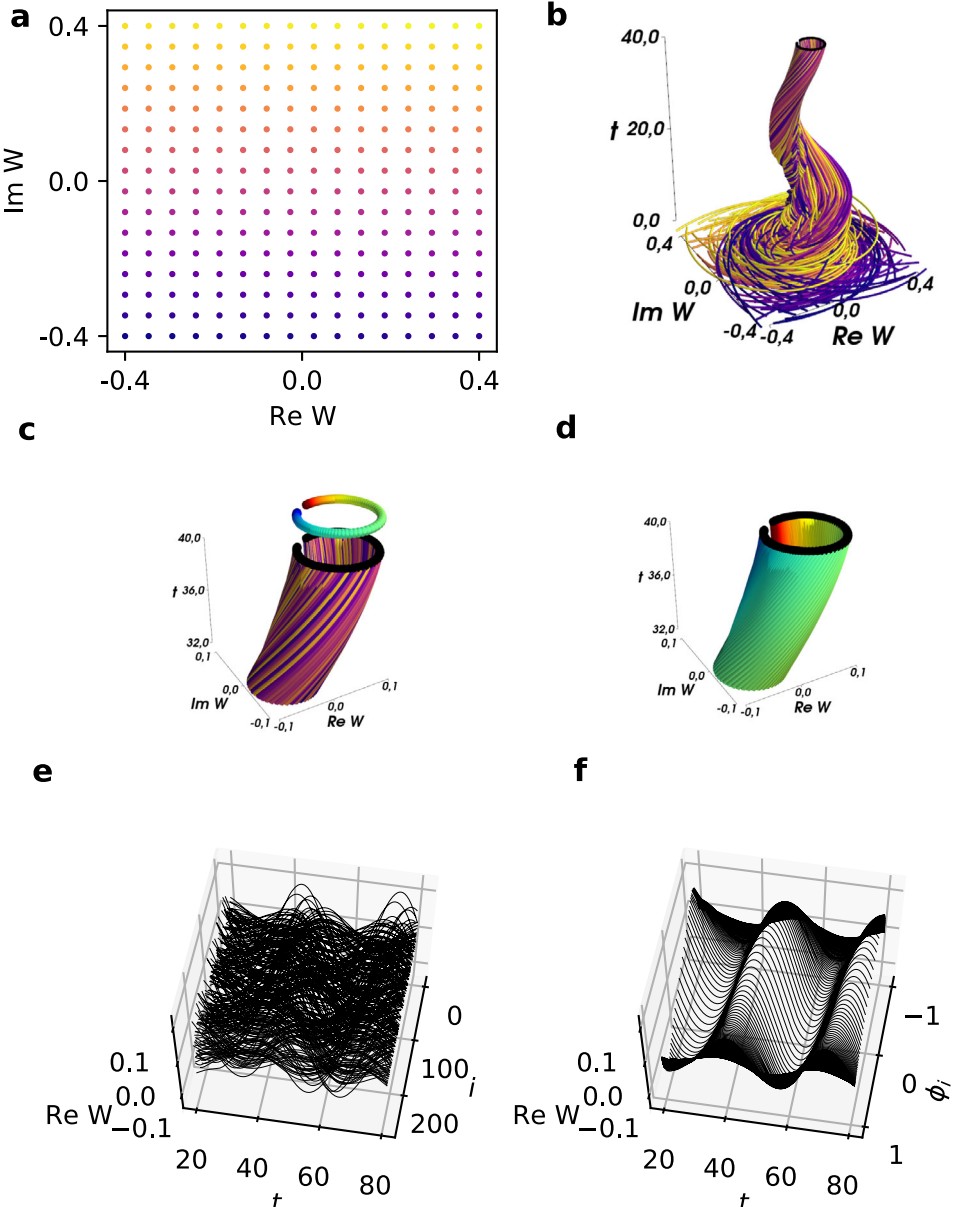

**Fig. 1 Illustration of the emergent PDE approach for coupled oscillator systems. a** Initial condition of the Stuart-Landau ensemble, Eq. (1), colored with ascending imaginary part of $W_k$. **b** Trajectories obtained from integrating the initial conditions of (**a**) with the same color coding as in (**a**). The last snapshot is marked by black dots. **c** Zoom in to the upper part of (**b**), with the last snapshot marked by black dots. Above it, the last snapshot is color coded based on the ordering of the oscillators along the curve at that moment. **d** Zoom in on the top part of (**b**), but now with the new color coding. **e** Trajectories of the real part of the $W_k$, arranged by their initial values Im$W$. **f** Trajectories of the real part of the $W_k$, arranged by the new color coding $\phi_i$ as in (**d**). (Finding $\phi_i$ is discussed in the text).

each oscillator $k = 1, \ldots, N$ is represented by a complex variable $W_k$ and coupled to all other oscillators through the ensemble average. The long-range interaction is in fact global, since the coupling is all-to-all. Each agent, when uncoupled, undergoes periodic motion with its own intrinsic frequency $\omega_k$, different across agents, making the ensemble heterogeneous.

Suppose we initialize an ensemble of $N = 256$ oscillators with values $W_k$ on a regular grid, as shown in Fig. 1(a). The color coding thereby correlates with the imaginary part of $W_k$. Integrating this initial condition using Eq. (1) with coupling constant $K = 1.2$ and intrinsic frequencies $\omega_k$ distributed equally spaced within the interval $[-1.5, 1.9]$ yields the dynamics in Fig. 1(b): although the behavior appears quite irregular at the beginning, it quickly settles onto a cylinder-like structure. Note that the color

coding is still the same. After the transients decay, the agents appear arranged on this structure in an irregular manner if colored based on their initialization, see the zoom in of the upper part as shown in Fig. 1(c). Using manifold learning, we will show that it is possible to find a parametrization of the agents (a different coloring) in which the dynamics appears more ordered and regular. This is shown by the new color coding of the last snapshot in Fig. 1(c), and the recolored attractor in Fig. 1(d). Indeed, when contrasting the time series of the agents in the original color coding $i$ (Fig. 1(e)) and the new color coding $\phi_i$ (Fig. 1(f)), we argue that the dynamics appear more regular in a space parametrized by $\phi_i$, suggesting the possibility that the solution can be described by a PDE with $\phi_i$ and time as the independent variables.

The remainder of this article is organized as follows: First, we illustrate our approach through a caricature, where we start with a known PDE in a predefined spatial variable. We observe the dynamics at a number of mesh points in this known space, but then we scramble the time series ourselves, on purpose, concealing the spatial coordinates of where the behavior was observed. We obtain a predictive PDE description in a learned emergent spatial or heterogeneity coordinate $\tilde{x}$, discovered through data mining these scrambled behaviors. We then confirm that this emergent coordinate is one-to-one with the (discarded) physical location $x$ of the original mesh points.

Returning to our globally-coupled oscillator ensemble, we show how to extract an intrinsic space coordinate, and learn a PDE description in this parametrization and time. We then study parametric dependencies of this PDE: we sample dynamics at parameter values bracketing a (collective) Hopf bifurcation. Using this data, we show that learning a PDE with an additional input for a parameter can capture the location and nature of bifurcations in this parameter.

We then go beyond a single emergent space dimension: For a biologically-motivated mathematical model of coupled Hodgkin-Huxley type neurons, used to describe the dynamics in the pre-Bötzinger complex of the brain, data mining discovers that the description of the agent behaviors is now two-dimensional. We again learn a PDE describing the agent dynamics - now in two emergent space coordinates and time.

We conclude with a discussion of the approach and its shortcomings, and what we perceive as open questions and directions for future research. We also discuss the explainability of the learned emergent coordinate(s) for such agent-based systems. Details on the algorithms and numerical methods are summarized in the Methods section. The code to reproduce the results is available under https://github.com/fkemeth/emergent_pdes.

## Results

**Learning partial differential equations in emergent coordinates.** For an illustrative caricature, we use a PDE with a known independent space variable, before returning to our coupled agent example. In this case we do have a known independent spatial coordinate, $x$, but we will randomly scramble it ourselves, to validate that our algorithms can, in a meaningful way, recover it. Consider the 1D complex Ginzburg-Landau equation, a PDE for the evolution of a complex field $W(x, t)$ in one spatial dimension $x \in [0, L]$, defined by

$$\frac{\partial}{\partial t} W(x, t) = W(x, t) + \left(1 + ic_1\right) \frac{\partial^2}{\partial x^2} W(x, t) - \left(1 - ic_2\right) |W(x, t)|^2 W(x, t) \tag{2}$$

with real parameters $c_1 = 0$, $c_2 = -3$, $L = 80$, and, here, periodic boundary conditions. We integrate this system using a pseudo-spectral method with exponential time stepping[25]. This results in spatiotemporal chaotic dynamics, so called spatiotemporal intermittency, with the spatiotemproal evolution shown in Fig. 2(a). See section Methods for an additional example with $c_1 = 1$, $c_2 = 2$ and no-flux (Neumann) boundary conditions showing periodic dynamics.

For integration, the spatial coordinate $x$ is discretized into $N = 256$ equidistant points $x_k$. Eq. (2) thus yields $N$ (here complex) time series $W_k(t)$ at each mesh point $x_k$. We can think of the behavior at each mesh point as the behavior of an agent in an ensemble of interacting agents. Assuming the $x_k$ label of each agent is not available (cf. Fig. 2(b), where the agents are parametrized by a random index $i$); is it possible to find a collective description of the dynamics in these time series based

on a data-driven, emergent spatial variable, and in the form of a partial differential equation, involving partial derivatives in this variable?

We accomplish this by extracting an intrinsic independent coordinate from the time series data. As proposed in Ref. [9] we use diffusion maps (each of the scrambled time series is a data point) to extract coordinates parametrizing the ensemble of time series, see Methods. It may be qualitatively helpful (even though we use a nonlinear manifold learning algorithm) to think of this as performing principal component analysis (PCA) on the ensemble of time series (each of them is a data point) and then keeping the leading PCA component as an emergent spatial coordinate. This emergent coordinate is used to parametrize a useful embedding space in which to learn a PDE.

For the time series data in Fig. 2(b), we find two independent diffusion modes $\phi_1$ and $\phi_2$, spanning a circle in diffusion maps space, which is shown in Fig. 2(c). This circle is one-to-one with the original periodic domain, however, through the scrambling, the time series $W_k$ are located at random position along this circle (see color coding in Fig. 2(c)). Even without knowledge of the spatial location of the mesh points, we can still extract a data-driven coordinate $\tilde{x}$ parametrizing the circle (see color coding in Fig. 2(d)), and set out to learn a PDE with this coordinate as the spatial dimension. The data parametrized this way is depicted in Fig. 2(e). Note that $\tilde{x}$ is one-to-one with, but not identical, to $x$. In particular, it is shifted (see the shifts in Figs. 2(a) and (e)) due to the non-uniqueness of the parametrization of the periodic domain. We now set out to learn a PDE description based on partial derivatives in $\tilde{x}$,

$$\frac{\partial}{\partial t} W(\tilde{x}, t) = f\left(W, \frac{\partial W}{\partial \tilde{x}}, \frac{\partial^2 W}{\partial \tilde{x}^2}, \frac{\partial^3 W}{\partial \tilde{x}^3}\right) \tag{3}$$

where $f$ is represented by a fully connected neural network. See Methods for details on the neural network architecture and the data sampling. A number of issues arise in learning such a PDE in $\tilde{x}$:

- Since $\tilde{x}$ is in general not identical to $x$, trajectories $W_k$ are not equally spaced. To calculate a finite difference approximation of $\partial^n W / \partial \phi_1^n$, we interpolate the $\tilde{x}$-parametrized data using cubic splines and sample $W$ at $N = 256$ equidistant points on the interval $[-\pi, \pi]$.
- PDEs define properties of functions in infinite dimensional spaces; we cannot sample the full state space, and so our learned surrogate PDE will not know the dynamics in all state space directions. Various techniques proposed in recent years (especially in imitation learning) attempt to regularize surrogate dynamical systems. These include contraction theory[26–29], and convex neural networks[30,31]. They rely on the existence of a Lyapunov function; other approaches include Jacobian regularization[32,33]. However, they usually involve additional loss terms or are computationally expensive.

Here, we sample multiple transients towards the attractor as training data, and, if necessary, regularize the output of the learned PDE as follows: Using the simulation data, we create a truncated singular value decomposition (SVD) based on all the sampled transients. During inference, we filter the state obtained by integration of the neural network output by projecting it back onto this truncated SVD subspace, thus keeping the predicted trajectories there.

Integrating from an initial snapshot using the learned PDE $f$ in the emergent variable $\tilde{x}$ is shown in Fig. 2(f). Notice the close correspondence between predicted and actual dynamics, cf. Fig. 2(e).

In the next Section, we will follow the same approach, but now for a system where there is no original space coordinate to recover.

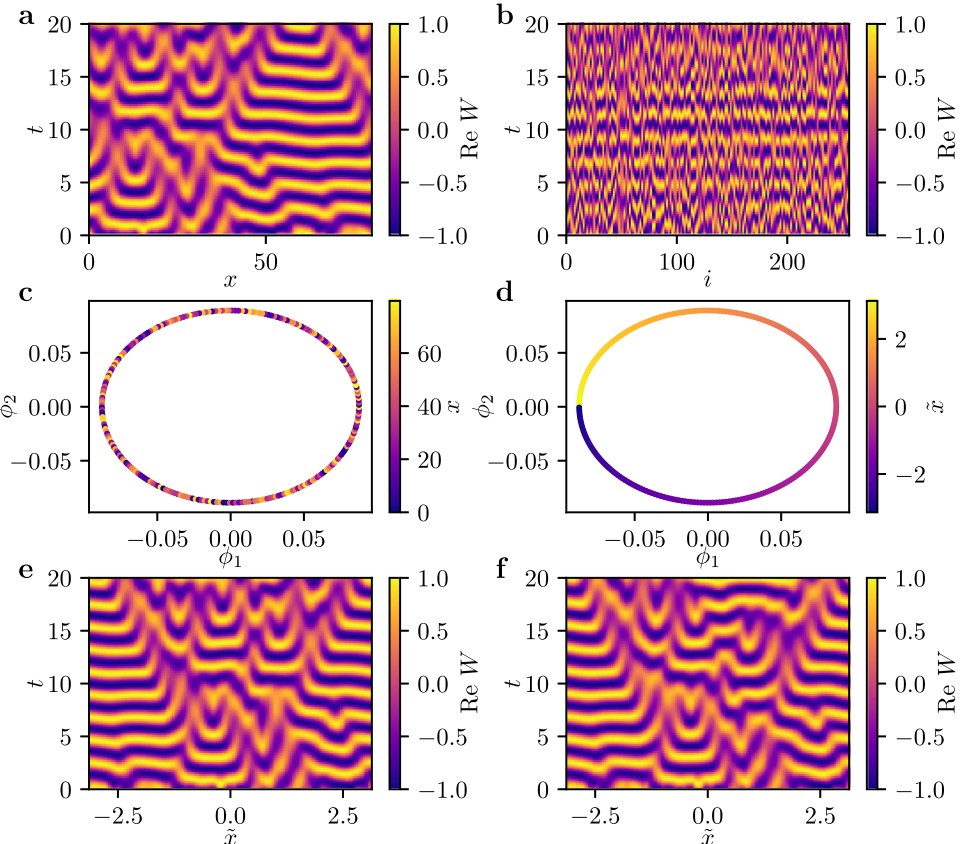

**Fig. 2 Data-driven PDE for the chaotic dynamics in the complex Ginzburg-Landau equation. a** The real part of the complex field $W(x, t)$ obtained from simulating Eq. (2) with $N = 256$ mesh points after initial transients have decayed. **b** Removing the spatial label yields a collection of $N$ time series plotted here in random sequence. **c** Using manifold learning (here diffusion maps), one finds that there exists two modes $\phi_1$ and $\phi_2$ parametrizing these time series. Each point corresponds to one of the $N$ time series, and is colored by its scrambled spatial location $x$. **d** Having obtained the embedding, we can introduce an emergent coordinate $\tilde{x}$ parametrizing the circle spanned by $\phi_1$ and $\phi_2$. **e** The real parts of the time series parametrized by $\tilde{x}$. **f** Real part of simulation predictions for the complex variable $W$ starting from an initial condition in our test set, using the partial differential equation model learned with $\tilde{x}$ as the spatial variable and a periodic domain.

**Learning Partial Differential Equations for Coupled Stuart-Landau Oscillator Dynamics**. Recall the original problem, Eq. (1), of an ensemble of mean-coupled Stuart-Landau oscillators,

$$\frac{\mathrm{d}}{\mathrm{d}t} W_k = (1 + i\omega_k) W_k - |W_k|^2 W_k + \frac{K}{N} \sum_j (W_j - W_k) \quad (4)$$

with $k = 1, \ldots, N$ and the real coupling constant $K$. The intrinsic frequencies $\omega_k$ are taken linearly spaced in the interval $[-\gamma + \omega_0, \gamma + \omega_0]$. Depending on the parameters $K$ and $\gamma$, a plethora of different dynamical phenomena are known to arise. Examples range from frequency locked oscillations and quasiperiodic dynamics to chaos and oscillator death. See Ref. [34] for a more detailed discussion. Here, we fix $K = 1.2$, $\gamma = 1.7$ and $\omega_0 = 0.2$ - resulting in periodic, synchronized oscillations: the oscillators in the ensemble oscillate with a common frequency and maintain a constant mutual phase difference. The real part of such dynamics is depicted in Fig. 3(a), parametrized by $\phi_1$, the first independent diffusion map mode. As for the complex Ginzburg-Landau equation, we sample data not only on the attractor, but also on transients in its neighborhood approaching it. These long-term dynamics can be thought of as lying on an attracting slow manifold; see Methods.

The predictions from an initial condition on the limit cycle using the learned PDE model are depicted in Fig. 3(b), and closely resemble the actual dynamics, as depicted in Fig. 3(a). Note here that due to the deformation of the space coordinate, the boundary

conditions in the transformed variable may no longer be obvious. We therefore learn $f$ only in the interior of the $\phi_1$ domain. When we simulate the learned PDE, we provide (as boundary conditions) a narrow space-time data corridor as needed. The imposition of such finite corridor boundary conditions is particularly important for such agent-based systems as considered here, where the form of effective boundary condition formulas (like Dirichlet, Neumann or Robin) in the emergent space is not known a priori. The model also captures the dynamics approaching the limit cycle. This can be visualized by integrating from initial conditions on the slow manifold but off the attracting limit cycle. We integrated such an initial condition from our test set using forward Euler and both the full ODE system, Eq. (1), as well as the learned emergent PDE model. The smallest Euclidean distance in $\mathbb{C}^N$ between these transients and the true attractor at each time step is depicted in Fig. 3(c). Note that both the true and learned transients converge to the limit cycle at a similar rate, and the learned PDE trajectory approximates the behavior of the full ODE system well. In an attempt to obtain a physical meaning of the emergent coordinate $\phi_1$, we plot it as a function of the intrinsic frequency $\omega$ of the oscillators in Fig. 3(d). It becomes obvious that the two quantities are one-to-one, analogous to the $(\tilde{x}, x)$ pair in the complex Ginzburg-Landau example above: our data mining has discovered the heterogeneity of the ensemble, and uses it to parametrize the dynamics. Knowing the equations and how $\omega_k$ enters in them, one could analytically attempt to

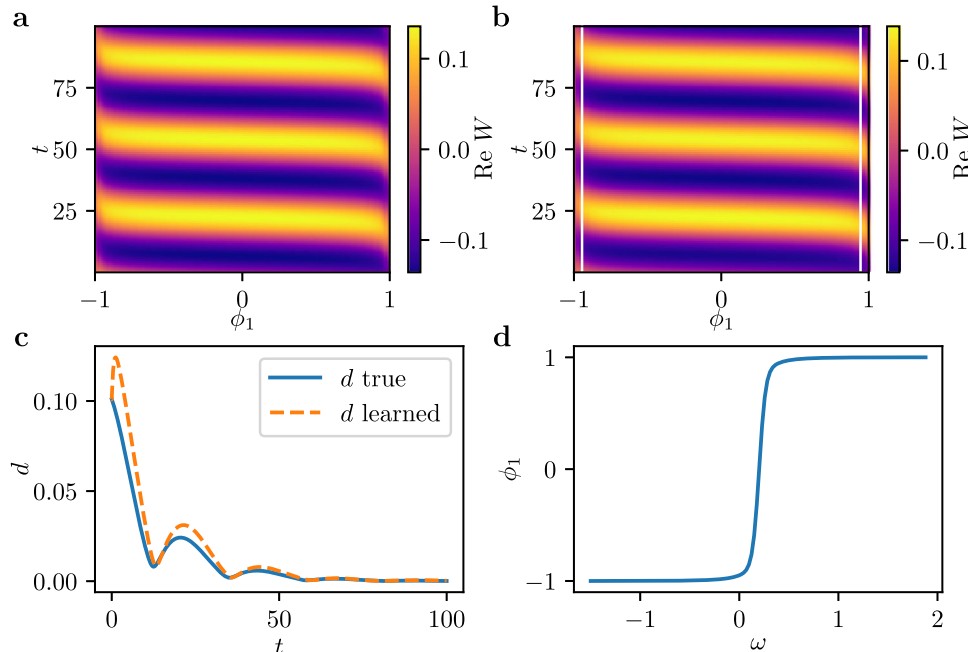

**Fig. 3 Emergent PDE for an ensemble of Stuart-Landau oscillators. a** Real part of the complex variable $W$ for a system of $N = 512$ oscillators, parametrized by the first emergent diffusion mode $\phi_1$. **b** Dynamics obtained from the learned model by integrating starting from the same initial snapshot as in (**a**). **c** Smallest Euclidean distance $d$ in $\mathbb{C}^N$ at each time step between the transients and the true attractor for the true PDE (blue) and the learned PDE (orange). **d** The first diffusion mode $\phi_1$ as a function of the intrinsic frequencies $\omega$ of the oscillator ensemble.

derive Ott-Antonsen-type equations (for phase oscillators) in $\omega$ space[3]. We know neither the equations, nor the $\omega_k$ (and the oscillators are not phase oscillators to boot); everything here is data-driven.

Having been successful in capturing the attractor and its nearby dynamics for a single parameter value, it becomes natural to explore whether the learned PDE can also capture bifurcations: qualitative changes in the dynamics when changing system parameters. In particular, for $\gamma = \gamma_H \approx 1.75$, the Stuart-Landau ensemble undergoes a collective Hopf bifurcation, at which the amplitude of the oscillations shown in Fig. 3 vanishes. For $\gamma > \gamma_H$, a stable fixed point ensues, in which all individual amplitudes of the respective oscillators are zero, also called oscillator death[35]. We now collect data for training at several $\gamma$ values, linearly spaced in the interval [1.7, 1.8], on both sides of the Hopf bifurcation; the $\gamma$ value was provided as additional input to the model. We again perturbed along the slow stable eigendirections of each attractor, see Methods, collecting transients that inform the model about nearby dynamics. We then learned a PDE of the form

$$\frac{\partial}{\partial t} W(\phi_1, t) = f\left(W, \frac{\partial W}{\partial \phi_1}, \frac{\partial^2 W}{\partial \phi_1^2}, \frac{\partial^3 W}{\partial \phi_1^3}; \gamma\right). \tag{5}$$

The learned dynamics, starting from an initial oscillator ensemble profile, and integrated using the learned model are shown in Fig. 4 for $\gamma < \gamma_H$ (left inset) and for $\gamma > \gamma_H$ (right inset). We observe the transient dynamics approaching the fixed point $W = 0 \,\forall\, \omega$ for $\gamma = 1.8$.

Validating the approach further, we start at random initial conditions in the slow eigenspace of the attractor at different $\gamma$ values using the Stuart-Landau system, Eq. (1), as well as the learned PDE model. For both models, we record a snapshot after $T = 10000$ dimensionless time units and calculate its average amplitude $\langle|W_{\text{limit}}|\rangle$. An average amplitude equal to zero then indicates that the initial condition converged to the fixed point $W = 0 \,\forall\, \omega$ under the respective model, whereas a nonzero

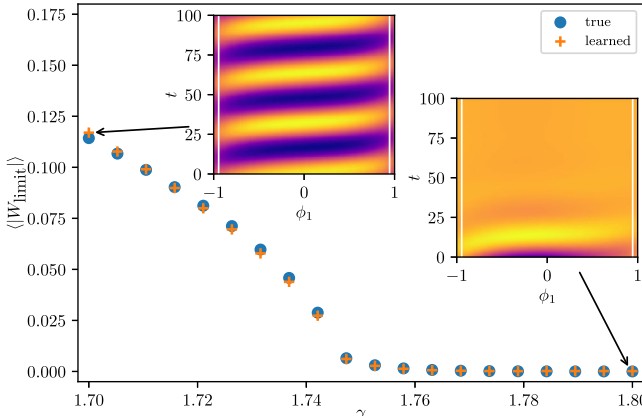

**Fig. 4 Computational bifurcation diagram by plotting the mean amplitude $\langle|W_{\text{limit}}|\rangle$ averaged over the ensemble at the limit set.** In particular, we integrate from random initial conditions close to the limit set for $T = 10000$ dimensionless time units for the Stuart-Landau ensemble (blue circles) and the learned PDE (orange crosses). A mean amplitude near zero indicates convergence to the fixed-point $W = 0 \,\forall\, \omega$, whereas a non-zero $\langle|W_{\text{limit}}|\rangle$ indicates oscillations with finite amplitude. The color codings of the insets show the real part of the complex variable $W$ obtained from integrating an initial condition close to the fixed point $W_k = 0$ with $\gamma = 1.8$ (right inset) and close to the limit cycle with $\gamma = 1.7$ (left inset) using the learned model and employing explicit forward Euler for $\gamma = 1.8 > \gamma_H$.

amplitude indicates convergence to the (collective/spatiotemporal) limit cycle. The resulting $\langle|W_{\text{limit}}|\rangle$ values for different $\gamma$ are shown in Fig. 4, with blue circles for the original dynamics and orange crosses for the learned dynamics. The Hopf bifurcation manifests itself in the sudden increase in amplitude when $\gamma$ is varied. Note the close correspondence between the learned model and the original oscillator system: both converge to

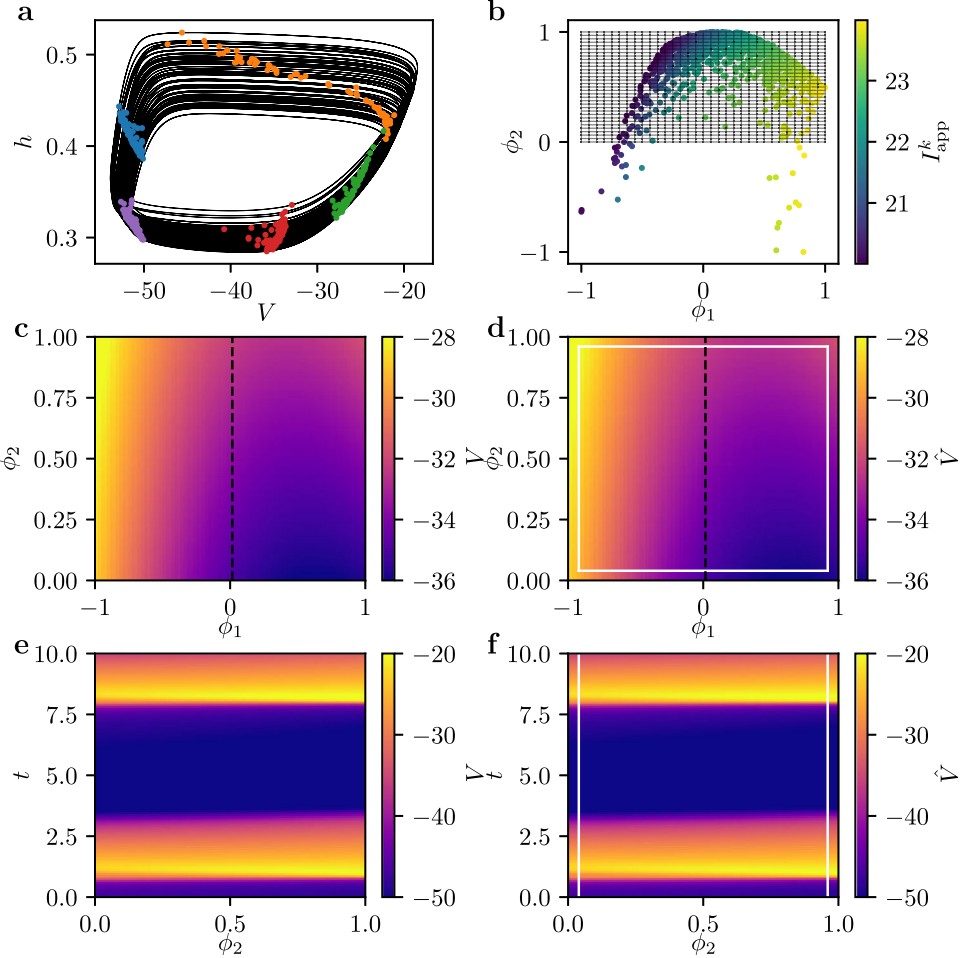

**Fig. 5 Emergent PDE for a network of Hodgkin-Huxley like neurons in the pre-Bötzinger complex. a** Trajectories of the ensemble and five snapshots (colored points) in the $V, h$ plane of an ensemble of 1024 neurons. For better visability, only 64 trajectories are shown. **b** The two emergent coordinates $\phi_1$ and $\phi_2$. Through the color coding with the intrinsic heterogeneity $I_{app}^k$, one can observe that $I_{app}^k$ is a function of the emergent space coordinates. The rectangular grid indicates the space in which we chose to learn an effective PDE. **c** Snapshot of $V$ at $t = 10$ obtained by fitting the simulation data on the grid shown in (**b**). **d** Snapshot of $V$ at $t = 10$ predicted by the learned PDE model. **e** Space-time plot of the evolution of $V$ at the cut $\phi_1 = 0$, as indicated in (**c**). **f** Predictions $\hat{V}$ of the space-time evolution of $V$ at $\phi_1 = 0$. The white lines indicate the borders of the boundary conditions.

a fixed point for $\gamma > \gamma_H \approx 1.75$, and to the limit cycle for $\gamma < \gamma_H \approx 1.75$.

**Two emergent spatial coordinates**. The approach can easily be extended to situations with more than one emergent spatial dimension, that is, to problems in which more than one diffusion map component become necessary to parametrize the inherent heterogeneity of agent behaviors. As an example, we consider a system of coupled Hodgkin-Huxley type neurons, a caricature for modeling the dynamics in the pre-Bötzinger complex[36–38]. The state of the $k$-th neuron (out of 1024 total neurons) is specified by a channel variable $h_k$ and a voltage variable $V_k$. In addition, the neurons are coupled such that they form a random Chung-Lu type network. This means that the number of connections of each neuron varies from neuron to neuron. Furthermore, the neurons differ in the value of the kinetic parameter $I_{app}^k$ in the equations. Thus, the model has two heterogeneous parameters: a structural heterogeneity resulting from the network topology and an intrinsic heterogeneity through the applied current $I_{app}^k$. See the Methods section for details on the dynamical equations of the model.

Fig. 5 (a) depicts the dynamics of the model for $N = 1024$ neurons. The black lines indicate trajectories of a subset of these

neurons, whereas the colored dots mark snapshots. Note that the system is periodic in time but the neurons are spread out at each time step.

In Fig. 5(b), the emergent coordinates for such a dynamics are shown, obtained by performing diffusion maps on the collection of simulated time series. Note that there are two independent directions, $\phi_1$ and $\phi_2$, parametrizing the neurons. By coloring $\phi_1$ and $\phi_2$ with the intrinsic heterogeneity $I_{app}^k$, one can observe that one emergent space direction correlates with this parameter. One can furthermore show that the second direction approximately corresponds to the connectivity degree of each neuron in the network, the number of other neurons it is directly connected to[9].

Our contribution in this paper is to learn an effective PDE in a rectangular interval in emergent space, as indicated by the grid shown in Fig. 5(b). This is achieved by fitting polynomials to the data and interpolating on the regular grid points, see Methods. One snapshot of $V$ at $t = 10$ is depicted in Fig. 5(c). Using the interpolated data along the attractor and of a few transients, we learn a PDE as described in the previous sections. However, the input to the neural network now consists of partial derivatives of the $h$ and $V$ fields with respect to $\phi_1$ and $\phi_2$, obtained using finite differences. One can then use the model to predict the dynamics of a so far unseen initial snapshot. A snapshot of $V$ at $t = 10$

obtained by integrating the same initial condition as in Fig. 5(c) using the learned PDE and forward Euler is shown in Fig. 5(d). The white lines thereby indicate the extent of the thin boundary corridors provided in lieu of boundary conditions during integration. In Fig. 5(e), the space-time dynamics of $V$ along the one-dimensional cut $\phi_1 = 0$, as indicated by the dashed line in Fig. 5(c), is shown. The predicted dynamics of $V$, $\hat{V}$, along the same cut in emergent space is depicted in Fig. 5(f). Notice the close correspondence between the actual dynamics and the predictions of the learned model.

## Discussion

We have seen that it is possible to learn a predictive model for the dynamics of coupled agents based on local partial derivatives with respect to one (or more) emergent, data-driven spatial variable(s) and time, that is, in the form of a partial differential equation. As an example, we investigated an ensemble of mean-coupled Stuart-Landau oscillators, where each oscillator has an intrinsic frequency $\omega_k$. Using manifold learning (here, diffusion maps), we were able to extract an intrinsic coordinate $\phi_1$ from time series segments of these oscillators. Starting with just a single parameter value $\gamma = 1.7 < \gamma_H$, our results indicate that a model based on a few partial derivatives with respect to $\phi_1$ is able to accurately capture the collective dynamics in the slow manifold and on the final attracting limit cycle. These results extend to the case in which data is sampled for different $\gamma$ values on both sides of the Hopf bifurcation point $\gamma_H$. The learned PDE then modeled successfully the slow transients towards either the stable limit cycle or the stable fixed point, depending on the parameter. We then extended our analysis to a biologically-motivated example where the agents are Hodgkin-Huxley type neurons. There, we found a two-dimensional embedding of the time series, and subsequently learned a PDE in this two-dimensional emergent space.

For a successful implementation of our approach we employed a systematic way of sampling training data: From a given limit set, we perturb along the slow stable manifold, and sample transients approaching the attractor. This sampling strategy is assisted by estimates of the slow stable directions (and their time scales) through the linearized system Jacobian, that help produce informative initial conditions. Because of the fast-slow nature of the dynamics, we found that starting practically anywhere and integrating for a short time will bring the dynamics close to this slow manifold.

This ought to also be the case when collecting experimental data (discarding short initial transients to the slow manifold). Clearly, the model cannot be expected to learn the right asymptotic behavior in dimensions in which it has seen no data. This can lead to instabilities when attempting to predict the long term dynamics of the system. We addressed this problem through filtering, in particular through a truncated SVD regularization. An SVD basis was constructed from the training data, and, during inference, we filtered by projecting the predictions on this basis; the predicted dynamics cannot leave the space spanned by the truncated SVD. This introduces an additional hyperparameter to the model: the dimension after which to truncate the SVD used for filtering. Too many dimensions may allow for instability in the predictions (lack of training data); too few leads to poor representations and distorted dynamics. Our threshold was empirically chosen by trial and error; the subject is worthy of a more detailed study. Other approaches may be employed as well, such as hyperviscosity in the learned PDE model[39–41], effectively damping higher frequency components.

An important question in deciding which PDE model to learn, is how many emergent spatial derivatives one has to include in the PDE right hand side. In other words, how can one decide when $\partial W/\partial t$ is well approximated by $W$ and its derivatives with respect to $\phi_1$? For Gaussian process regression, recent work using Automatic Relevance Determination helps tackle this problem[42]. In our case we again decided empirically, by trial and error; a more thorough study must clearly follow. In addition, the issue of boundary conditions in emergent space (here we used narrow boundary corridors), as well as what constitutes a well posed problem for an operator identified in a data-driven way constitute important (and challenging) questions to pursue; we mention here the possibility of using the approach of the baby-bathwater scheme in[43].

Fig. 4(b) indicates that the learned model captures qualitative changes in the dynamics when changing a system parameter, here a Hopf bifurcation from a fixed point for $\gamma > \gamma_H$ to collective oscillations for $\gamma < \gamma_H$. More quantitatively, we reported the leading spectrum of the linearization of the model evaluated at the fixed point. This was obtained using automatic differentiation of the neural network model with respect to its inputs. Such computations can shed more light on the similarities and differences of agent-based simulations and their emergent PDE descriptions. In this paper, we focused on a particular regime in parameter space. However, our approach can easily be extended to more intricate dynamics that are known in such a Stuart-Landau ensemble; informative examples are included in the videos SI1 and SI2.

Historically, it is known that physical phenomena modeled at the fine scale through atomistic/stochastic/agent-based simulations are often well approximated using closed partial differential equations in terms of a few of their collective observables (e.g., moments of the particle distribution, such as the agent density). Our approach will be useful when we believe that such effective, collective PDE models in principle exist, but the closures required to write them down are not known. It can also provide useful results in regimes where the strong mathematical assumptions required to provably obtain explicit closures can be relaxed. This is an area where equation-free multiscale numerics has been used to solve the equations without writing them down, and where manifold learning has been used to even perform this solution (dependent) variable free, that is, in terms of dependent variables not known a priori, but revealed through data mining of detailed simulations (see, for example, the discussion in[44]). All scientific computation in latent space (e.g., see[45] and[46]) falls in this class.

What is different and exciting in the present study is the extension of this approach to problems where there are no obvious independent spatial variables - dynamics of coupled oscillators, dynamics on and of networks, dynamics of systems of interacting systems, where the right space for modeling the problem is not known a priori. Writing models in such an emergent activity space, with emergent space and even emergent time[9] coordinates may become a useful method for the modeler: a tool that extends the toolkit for linking domain science knowledge at the detailed level with machine/manifold learning to build useful, predictive models.

Here, we chose a model based on local descriptors, local in the emergent space. One can speculate about contexts in which such a local description might be beneficial. It certainly is more humanly parsimonious/compact to write down than the detailed list of all units and all interactions. It may also be convenient if one needs to make predictions with limited memory (limited fast cpu memory so to speak). We do not need to know what every unit is doing - we look at the activity of similar units (that are already embedded nearby in emergent space) and make predictions based on smoothness (mathematically expressed through

Taylor series) and the behavior of the neighbors. Our emergent space can then be thought of as a space where nearby (observations of) behaviors come already usefully clustered. Alternatively, we can think of this space as embodying a useful attention geometry - the behaviors we need to pay attention to (because of their similarity) in order to make a prediction, are already our neighbors in this space. Geometric proximity in the emergent space saves us then from having to search for comparable behavior histories across all interacting units in physical space-time. This enables us to exploit smoothness across behavior histories in order to make local predictions with only a few nearby data. In our Stuart-Landau example, the oscillators are globally coupled, while we find a local PDE (without integral terms) that successfully describes their behavior. This apparent disconnect of local PDE description versus global coupling can be explained through infinite propagation speed of information for certain parabolic PDE, such as the heat equation. Modeling globally coupled oscillators with a PDE that only allows finite propagation speed, such as the wave equation, would not lead to the correct behavior. In our case, the network automatically learned that infinite propagation speed is necessary, and we are still investigating how such qualitative behavior can be learned more effectively.

We touched briefly upon the explainability of our emergent spatial coordinates by showing that our $\phi_1$ was one-to-one with, and thus calibratable to, the oscillator intrinsic frequencies - the agent heterogeneity of the Stuart-Landau ensemble. In the Hodgkin-Huxley neuron example the emergent coordinates were again seen to be one-to-one with a parametrization of the oscillator heterogeneity; one corresponded approximately to the kinetic heterogeneity, while the second corresponded to the structural (connectivity) heterogeneity. The suggested approach then is to (a) decide how many emergent independent variables are necessary; (b) ask a domain scientist for physical quantities that may explain them and then (c) to test whether the explainable and the data-driven parametrizations are one-to-one on the data (the determinant of the Jacobian of the transformation is bi-Lipschitz, bounded away from zero and from infinity, on the data, e.g.,[47–49]).

Clearly, the explainability of predictive, generative equations in terms of data-driven dependent and independent variables, and operators approximated through machine learning is a crucial endeavor - when and why will we decide we trust results when we understand the algorithms, but do not understand the mechanistic, physical steps underlying the observations of what we model? Will a different understanding arise in latent/emergent space - analogous, say, to describing operators in Fourier space rather than physical space, or studying control in Laplace space rather than state space? From flocking starlings to interacting UAV swarms, this promises to be an exciting playing field for contemporary modelers.

## Methods

**Diffusion maps**. Diffusion maps use a kernel function to weigh pairwise distances between data points [23,24], typically the Gaussian kernel

$$k(x, y) = \exp\left(-\frac{\|\mathbf{x} - \mathbf{y}\|^2}{\epsilon}\right) \quad (6)$$

with a predefined kernel scale $\epsilon$ and a Euclidean distance metric, which we adopt here. The data points x, y are, in our case, the $N$ time series, resulting in a $\mathbf{K} \in \mathbb{R}^{N \times N}$ kernel matrix. Row-normalizing this kernel matrix yields a Markov transition matrix, also called diffusion matrix, and its leading independent eigenvectors corresponding to the largest eigenvalues can be used to parametrize the data[50].

Note that the eigenvectors of the diffusion matrix correspond to the eigenfunctions of the Laplace operator on the data manifold. As such, eigenvectors that can be written as functions of other eigenvectors with larger eigenvalue appear in the eigendecomposition of the diffusion matrix. An important task when using

diffusion maps is to extract the independent eigenvectors that parametrize new directions in the data. A prominent tool for this task was developed in Ref.[50] and is based on performing local linear regression on the set eigenvectors. Here, we perform visual inspection of the first ten eigendirections to investigate which eigenvectors are harmonics and which eigenvectors represent new directions in the data. These independent diffusion eigenvectors are then scaled to the interval $[-1, 1]$ for better comparison.

**Complex Ginzburg-Landau equation with spatiotemporal chaotic dynamics.** Consider the complex Ginzburg-Landau equation

$$\frac{\partial}{\partial t} W(x, t) = W(x, t) + (1 + ic_1)\frac{\partial^2}{\partial x^2} W(x, t) - (1 - ic_2)|W(x, t)|^2 W(x, t) \quad (7)$$

in one spatial dimension $x$, in a domain of length $L$. We solve this equation using random initial condition with periodic boundary conditions and parameter values $c_1 = 0$, $c_2 = -3$ and $L = 80$ using a pseudospectral method with exponential time stepping[25]. We sample data after initial transients have decayed, i.e., after 1000 dimensionless time units. The subsequent spatiotemporal evolution is depicted in Fig. 2(a).

Data for training our model is sampled as described in the following: For the number of training examples, we set $n_{train} = 20$ and for the number of test examples $n_{test} = 2$, yielding $n_{total} = 22$. We thus integrate from random initial conditions $n_{total} = 22$ times for 1000 dimensionless time units. We subsequently perturb the resulting snapshot by adding again noise to the solution. In this way, we perturb off the attractor a bit, allowing our model to learn the stability of the attracting manifold. We then integrate each perturbed snapshot for another 20 dimensionless time units, and sample data every $dt = 0.02$ time steps. This means, in total there are 20000 snapshot data pairs for training, and 2000 for validation. In order to find a parametrization for the discretization points of the PDE, we concatenate the training time series of the $N = 256$ points, resulting in $20,000 \times 20$ long trajectories. Then, we use diffusion maps with a Euclidean distance and a Gaussian kernel, and take the kernel scale $\epsilon = 100$ such that only close time series effectively influence the diffusion maps calculation. This results in the two independent modes $\phi_1$ and $\phi_2$, as shown in Fig. 2(c). We then parametrize the circle by using the angle $\tilde{x} \in [-\pi, \pi[$. We resample data on a regular grid in the interval $[-\pi, \pi[$ using a cubic spline. We estimate the time derivative at each point using finite differences in time,

$$\frac{\partial}{\partial t} W(\tilde{x}, t_j) = \partial_t W(\tilde{x}, t_j) \approx (W(\tilde{x}, t_j + dt) - W(\tilde{x}, t_j))/dt. \quad (8)$$

Using the $(W(\tilde{x}, t_j), \partial_t W(\tilde{x}, t_j))$ pairs, we train a neural network $f$ in a supervised manner as follows: We take $N = 256$ discretization points on each snapshot. At these points we calculate the first $n_{derivs} = 2$ spatial derivatives using a finite difference stencil of length $l = 5$ and the respective finite difference kernel for each spatial derivative of the highest accuracy order that fits into $l = 5$. The model thus takes the form

$$\partial_t W(\tilde{x}_i, t_j) \approx f(W(\tilde{x}_i, t_j), \partial_{\tilde{x}} W(\tilde{x}_i, t_j), \partial_{\tilde{x}\tilde{x}} W(\tilde{x}_i, t_j)) \quad (9)$$

with the derivatives calculated in the emergent space coordinate $\tilde{x}$ as described above. Note that $W(\tilde{x}, t)$ is complex, which means at each $(\tilde{x}_i, t_j)$ the input to the neural network is 6-dimensional for $n_{derivs} = 2$. The network itself is composed of 4 fully connected hidden layers with 96 neurons each and Swish activation function (resulting in $\approx 28 \cdot 10^3$ trainable parameters). The output layer contains two neurons with no activation function, one neuron for the real and imaginary part of $\partial_t W$, respectively. The network weights are initialized uniformly using PyTorch's default weight initialization[51], and are optimized using the Adam optimizer[52] with initial learning rate of $2 \cdot 10^{-3}$ and batch size of 128. Mean-squared error between the predicted and actual $\partial_t W(\tilde{x}_i, t_j)$, Eq. (8), is taken as the loss. The model is trained for 400 epochs, and the learning rate reduced by a factor of 2 if the validation loss does not decrease for 10 epochs. Needless to say, other general purpose approaches to learning the right-hand-side of the operator (Gaussian Processes[42], Geometric Harmonics[53], etc.) can also be used.

Inference is done by taking an initial perturbed snapshot of the validation data and integrating it forward in time using the learned model by using Scipy's Runge-Kutta-4(5) method and again periodic boundary conditions. The results are depicted in Fig. 2(f).

**Complex Ginzburg-Landau equation with periodic dynamics.** Consider the complex Ginzburg-Landau equation

$$\frac{\partial}{\partial t} W(x, t) = W(x, t) + (1 + ic_1)\frac{\partial^2}{\partial x^2} W(x, t) - (1 - ic_2)|W(x, t)|^2 W(x, t) \quad (10)$$

in one spatial dimension $x$, in a domain of length $L$. We integrate starting with initial condition

$$W(x, 0) = \frac{1 + \cos\frac{x\pi}{L}}{2} \quad (11)$$

using a finite-difference method in space and an implicit Adams method for

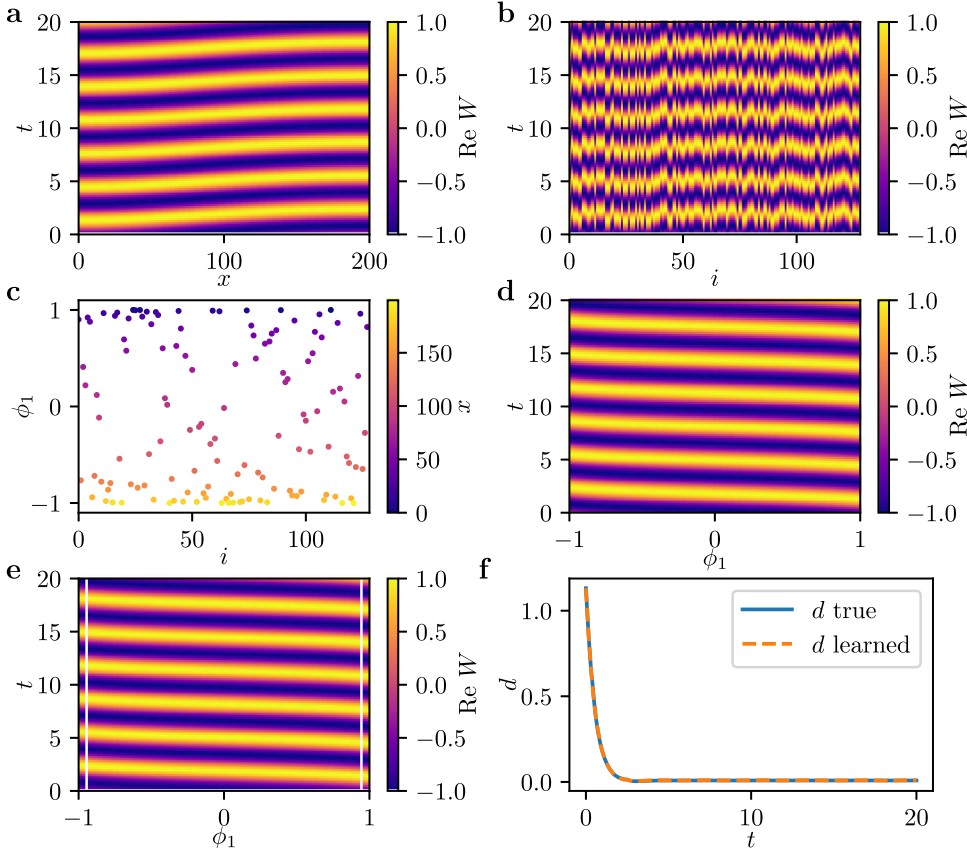

**Fig. 6 Data-driven discovery of the complex Ginzburg-Landau equation. a** The real part of the complex field $W(x, t)$ obtained from simulating Eq. (2) with $N = 128$ mesh points after initial transients have decayed. **b** Removing the spatial label yields a collection of $N$ time series plotted here in random sequence. (c) Using manifold learning (here diffusion maps), one finds that there exists a one-dimensional parametrization $\phi_1$ of these time series. Each point corresponds to one of the $N$ time series, and is colored by its actual spatial location $x$. **d** The real parts of the time series parametrized by $\phi_1$. **e** Real part of simulation predictions for the complex variable $W$ starting from an initial condition in our test set, using the partial differential equation model learned with $\phi_1$ as the spatial variable. Since no analytical boundary conditions are available, we provide the true values near the boundaries during integration, within a corridor indicated by white vertical lines. **f** Smallest Euclidean distance $d$ in $\mathbb{C}^N$ between the transients and the true attractor at each time step: true PDE (blue), learned PDE (orange).

integration, and sample data after initial transients have decayed, i.e., after 4000 dimensionless time units. This spatiotemporal evolution is depicted in Fig. 6(a).

We solve this equation using the initial condition

$$W(x, 0) = \left(1 + \cos\frac{\pi x}{L}\right)/2, \tag{12}$$

with zero-flux boundary conditions and parameter values $c_1 = 1$, $c_2 = 2$ and $L = 200$. It is worth noting that there is a slight left-right asymmetry in the solution shown in Fig. 6(a). Due to the symmetry of the space domain, there exist two stable solutions for this set of parameters; one has a slightly larger amplitude for large $x$, the other, reflected version, has a larger amplitude for small $x$. Choosing the initial condition defined above leads to a convergence to the same solution at every run. However, all initial conditions will eventually come down to a periodic solution.

Numerically, we integrate using a three point stencil for the finite difference approximation of the second derivative $\partial^2/\partial x^2$ with $N_{\text{int}} = 256$ discretization points and an implicit Adams method with $dt = 10^{-3}$ for the temporal evolution. The resulting behavior is depicted in Fig. 6(a). Data for training our model is sampled as described in the following: For the number of training examples, we set $n_{\text{train}} = 20$ and for the number of test examples $n_{\text{test}} = 1$, yielding $n_{\text{total}} = 21$. At $n_{\text{total}} = 21$ points along the limit cycle shown in Fig. 6(a), we sample data as follows: At $t_i = t_{\min} = 2000 + i d\tau$ with $i \in \{0, \ldots, n_{\text{total}} - 1\}$, with $d\tau = 100$, we perturb the limit cycle by scaling the respective snapshot at $t_i$ as $0.9 \cdot W(x, t_i)$ and $1.1 \cdot W(x, t_i)$. We integrate both of these snapshots forward in time for $T = 20$ time units, and sample data after each $dt = 10^{-3}$. This results in two transients, each comprised of 20,001 snapshots at each $t_i$. This means, in total there are $2 \times 20,000 \times 20 = 8 \cdot 10^5$ snapshot data pairs for training, and $2 \times 20,000$ for validation. We subsequently downsample the data to $N = 128$ points per snapshot. In order to find a parametrization for the discretization points of the PDE, we concatenate the training time series of the $N = 128$ points, resulting in $2 \times 20000 \times 20$ long trajectories. Then, we use diffusion maps with an Euclidean distance and a Gaussian kernel, and take the kernel scale $\epsilon$ as the median of all squared distances. This results in the one-dimensional parametrization $\phi_1$, as shown in Fig. 6(c). We resample data on a regular grid in the interval $[-1, 1]$ using a cubic spline. We

estimate the time derivative at each point using finite differences in time,

$$\frac{\partial}{\partial t} W(x, t_j) = \partial_t W(x, t_j) \approx (W(x, t_j + dt) - W(x, t_j))/dt, \tag{13}$$

yielding 20000 $(W(x, t_j), \partial_t W(x, t_j))$ pairs per transient and $t_i$.

Using the $(W(x, t_j), \partial_t W(x, t_j))$ pairs, we train a neural network $f$ such that

$$\partial_t W(x, t_j) \approx f(W(x, t_j)) \tag{14}$$

in a supervised manner as follows: We take $N = 128$ discretization points on each snapshot. At these points we calculate the first $n_{\text{derivs}} = 3$ spatial derivatives using a finite difference stencil of length $l = 9$ and the respective finite difference kernel for each spatial derivative of the highest accuracy order that fits into $l = 9$. The model thus takes the form

$$\partial_t W(x_i, t_j) \approx f(W(x_i, t_j), \partial_x W(x_i, t_j), \partial_{xx} W(x_i, t_j), \partial_{xxx} W(x_i, t_j)) \tag{15}$$

with the derivatives calculated as described above. Note that $W(x, t)$ is complex, which means at each $(x_i, t_j)$ the input to the neural network is 8-dimensional for $n_{\text{derivs}} = 3$. The network itself is composed of 4 fully connected hidden layers with 96 neurons each and tanh activation function (resulting in $\approx 28 \cdot 10^3$ trainable parameters). The output layer contains two neurons with no activation function, one neuron for the real and imaginary part of $\partial_t W$, respectively. The network weights are initialized uniformly using PyTorch's default weight initialization[51], and are optimized using the Adam optimizer[52] with initial learning rate of $10^{-3}$ and batch size of 1024. Mean-squared error between the predicted and actual $\partial_t W(x_i, t_j)$, Eq. (13), is taken as the loss. The model is trained for 60 epochs, and the learning rate reduced by a factor of 2 if the validation loss does not decrease for 7 epochs. Needless to say, other general purpose approaches to learning the right-hand-side of the operator (Gaussian Processes[42], Geometric Harmonics[53], etc.) can also be used.

Inference is done by taking an initial snapshot of the validation data near or on the limit cycle and integrating it forward in time using the learned model and an integration scheme such as forward Euler. At each time step, the boundary conditions (in the form of narrow boundary corridors) are taken from the ground-

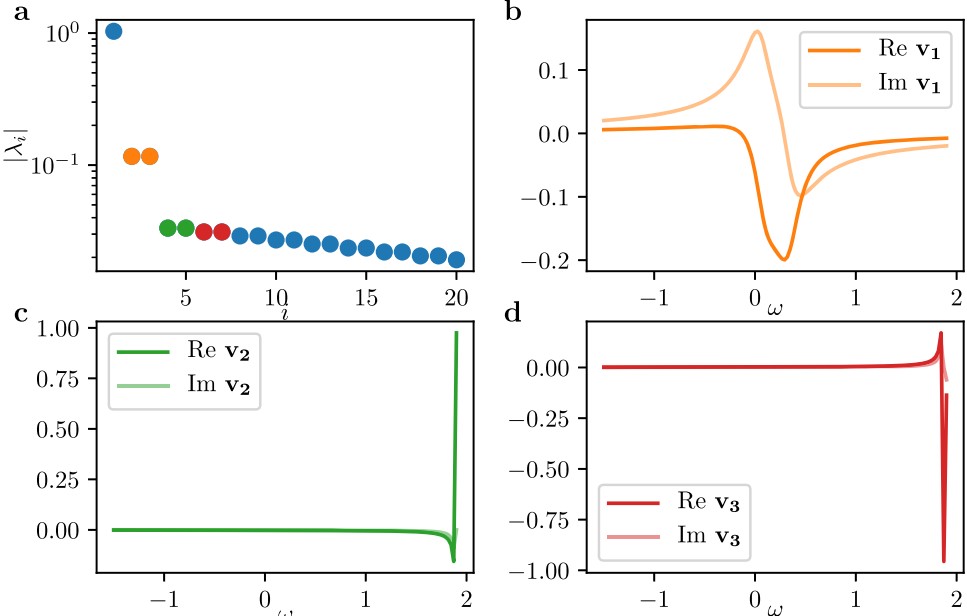

**Fig. 7 Floquet multipliers and corresponding eigendirections of the Stuart-Landau ensemble. a** Absolute values of the Floquet multipliers, $|\lambda_i|$, obtained from the monodromy matrix for the dynamics shown in Fig. 3. **b** Eigendirection $\mathbf{v_1}$ corresponding to the pair of complex conjugate multipliers $\lambda_2$ and $\lambda_3$ (marked in orange) indicating a slow attracting direction. **c**, **d** Eigendirections $\mathbf{v_2}$ and $\mathbf{v_3}$ corresponding to the pairs of complex conjugate multipliers $\lambda_4$, $\lambda_5$, and $\lambda_6$, $\lambda_7$, marked as green and red, indicating fast contracting directions. Note that since the $W_k$ are complex, the directions $\mathbf{v_i}$ are complex, with the real parts indicated as solid curves, and the imaginary parts indicated as shaded curves.

truth data. The issue arises of the right width for these corridors, and, more generally, the prescription of boundary/initial/internal conditions appropriate for the well-posedness of the overall problem, especially since the operator (the right hand side of the PDE) comes in the form of a black box. This is already the subject of extensive research that we, among others, are pursuing[54].

In addition, each predicted snapshot from the model is filtered as described in the following. On the whole training data set, an SVD is performed. Using the obtained $U$ and $V$ matrices, we can decompose each predicted snapshot during inference. In doing so, we truncate the SVD decomposition after two dimensions, and reconstruct the snapshot. This means that each snapshot is projected onto the two-dimensional subspace in which the training data lives, and thus prevents directions that have not been sampled from growing during inference. The resulting dynamics obtained from the learned model and using an initial snapshot on the limit cycle is depicted in Fig. 6(e). 4-point wide boundaries are provided on both sides of the domain. The learned dynamics can be investigated more clearly by comparing the true and the learned transient dynamics towards the limit cycle. To do so, we integrate a snapshot perturbed away from the limit cycle using the complex Ginzburg-Landau equation and the learned model, and calculate the smallest Euclidean distance in $\mathbb{C}^N$ at each time step of the obtained trajectories to the limit cycle. The results are shown in Fig. 6(f).

We also carefully checked that the learned model is converged with respect to the number of discretization points $N$.

**Stuart-Landau ensemble**. We integrate Eq. (1) using an implicit Adams method with the initial conditions of the oscillators uniformly distributed in the unit square in the complex plane. The intrinsic frequencies are thereby linearly spaced in the interval $[-1.5, 1.9]$, and the coupling constant is taken as $K = 1.2$. The dynamics as depicted in Figs. 1 and 3 are globally stable for the parameters considered here[34]. In fact, arbitrary initial conditions decay to the limit cycle exponentially. Such behavior can be investigated in more detail using Floquet theory: the convergence to the limit cycle can then be described by Floquet multipliers with their associate eigendirections. Since the limit cycle described above is stable, the absolute values of the Floquet multipliers are less than one, except for one of them which equals one. In particular, multipliers with large magnitude indicate slow attracting directions, whereas multipliers with absolute values close to zero indicate fast decaying directions. If both small and large Floquet multipliers are present, then there exist transients with multiple time scales. Following Ref. [55], we calculate the Floquet multipliers by calculating the monodromy matrix $\mathbf{V}$ along the limit cycle. In particular, we obtain $\mathbf{V}$ by the integration

$$\mathbf{V}(T) = \int_{t=0}^{t=T} \frac{\partial F}{\partial x}\Big|_{x(t)} \cdot \mathbf{V}\, dt \qquad (16)$$

with $\mathbf{V}(0) = \mathbf{I}_{2N \times 2N}$, $\mathbf{I}$ being the identity matrix, and $T$ being the period of one oscillation. The matrix $\frac{\partial F}{\partial x}$ represents the Jacobian of Eq. (1) obtained analytically

through differentiation and evaluated along the limit cycle. The eigenvalues of $\mathbf{V}(T)$ then correspond to the Floquet multipliers, with the corresponding eigenvectors being their respective directions.

The largest multipliers obtained this way, together with the three slowest eigendirections, are depicted in Fig. 7. Notice the single multiplier equal to one represents the neutral direction along the limit cycle. In addition, there is a pair of complex conjugate eigenvalues $\lambda_{2,3} \approx -0.4 \pm 0.4i$ (orange in Fig. 7). Due to the magnitude of their real parts, the dynamics in this eigenspace is slow compared to the subsequent eigendirections. These eigendirections are, as apparent from Fig. 7(b) smooth functions of the frequencies $\omega_k$. In addition, perturbations in this two-dimensional eigenspace spiral towards the stable limit cycle.

The directions of the subsequent multipliers affect only isolated oscillators. In particular, the subsequent direction (green in Fig. 7) following the slow eigenspace affects only the fastest oscillator, that is, the oscillator with the largest intrinsic frequency $\omega_k$. The next direction then perturbs the second fastest oscillator (red in Fig. 7), and so on. The step-like structure of the Floquet multipliers highlights the multi-scale behavior of the coupled oscillator system: The oscillation and the inward spiraling slow dynamics on one scale, and the single oscillator dynamics towards the limit on the other, the fast scale. These eigendirections with support on the most different oscillator are indicative of the SNIPER bifurcation marking the edge of synchronization.

We sample data by integrating system Eq. (1) from the random initial conditions described above, until the dynamics are settled on the limit cycle. For $n_{lc}$ different points along the limit cycle, we calculate the monodromy matrix from Eq. (16) and estimate the least stable eigendirection $\mathbf{v_1}$ transverse to the limit cycle, presumably lying on the slow stable manifold. Then, we perturb in this direction by perturbing each point $W_{lc}$ on the limit cycle as $W_{lc} \pm \epsilon \mathbf{v_1}$, with $\epsilon = 0.1$. This yields three initial points; integrating these points for a fixed amount of time then returns two transients towards the limit cycle and one trajectory on the attractor. Here, we choose $n_{lc} = 20$ for the training data, and $n_{lc} = 5$ for the test data, and a time window of $T = 200$ dimensionless time units with a sampling rate of $dt = 0.05$, yielding 4000 data points per trajectory, or $3 \cdot n_{cl} \cdot T/dt = 240{,}000$ training data points and 60,000 test data points. The concatenated time series of length $3 \cdot n_{lc} \cdot T/dt$ then serve as input data points for diffusion maps; the possibility of using time series snippets of different durations is explored in[9]. The temporal derivative $\partial_t W$ is then estimated using finite differences, cf. Eq. (13). When also changing the system parameter $\gamma$ we provide for each data point the corresponding $\gamma$ value as additional input to the network. In addition, the training data consists of uniform $\gamma$ values in $[1.7, 1.8]$, and the test data of randomly sampled $\gamma$ different from the training data. In addition, we estimate an SVD basis from the complete training data. During inference, the prediction of $f$ are reconstructed using this basis and a truncation with $n_s = 3$ dimensions.

For the extraction of diffusion modes, we use a kernel scale of $\epsilon = 20$ for the case when $\gamma$ is fixed and $\epsilon = 10$ when we sample data with different $\gamma$ values. Other hyperparameters and the model architecture are as described in the previous section.

**Heterogeneous network of Hodgkin-Huxley neurons**. Following Refs. [36–38], we model the dynamics of each neuron using the variables $V_k$ and $h_k$ as

$$C\frac{dV_k}{dt} = -g_{\mathrm{Na}} m(V_k) h_k (V_k - V_{\mathrm{Na}}) - g_l (V_i - V_l) + I_{\mathrm{syn}}^k + I_{\mathrm{app}}^k \quad (17)$$

$$\frac{dh_k}{dt} = \frac{h_\infty(V_k) - h_k}{\tau(V_k)}. \quad (18)$$

with $k = 1, \ldots, N$. The neurons are coupled through the synaptic current $I_{\mathrm{syn}}^k$ given by

$$I_{\mathrm{syn}}^k = \frac{g_{\mathrm{syn}}(V_{\mathrm{syn}} - V_k)}{N} \sum_{j=1}^{N} A_{kj} s(V_j) \quad (19)$$

with the symmetric adjacency matrix $A_{kj}$. The nonlinear functions $m(V)$, $h_\infty(V)$, $\tau(V)$ and $s(V)$ are given by

$$m(V) = \left(1 + \exp(-(V+37)/6)\right)^{-1} \quad (20)$$

$$h_\infty(V) = \left(1 + \exp((V+44)/6)\right)^{-1} \quad (21)$$

$$\tau(V) = \left(\epsilon \cosh((V+40)/5)\right)^{-1} \quad (22)$$

$$s(V) = \left(1 + \exp(-(V+40)/5)\right)^{-1} \quad (23)$$

with the constants $C = 0.21$, $\epsilon = 0.1$, $g_{\mathrm{Na}} = 2.8$, $g_l = 2.4$, $g_{\mathrm{syn}} = 0.3$, $V_{\mathrm{Na}} = 50$, $V_l = -65$, $V_{\mathrm{syn}} = 0$ and $N = 1024$. The applied currents $I_{\mathrm{app}}^k$ for each neuron $k$ are taken as $I_{\mathrm{app}}^k = 22 + 2\omega_k$ with $\omega_k$ being uniformly distributed in $[-1, 1]$.

The adjacency matrix $A_{kj}$ is constructed using the Chung-Lu network topology. Its entries are 1 with probability

$$p_{kj} = p_{jk} = \min\left(\frac{w_k w_j}{\sum_l w_l}, 1\right), \quad (24)$$

with $j < k$, and the weights $w_k$ being defined as $w_k = pN(k/N)^r$, $p = 0.9$, $r = 0.25$. Note that we take $A_{jk} = A_{kj}$ such that the adjacency matrix is symmetric.

We integrate the model using the Runge-Kutta method of order 5(4)[56] starting from identical initial conditions $V_k = -60$ and $h_k = 0$. We collect data after $t_{\min} = 120$ every $dt = 2 \cdot 10^{-3}$ time steps, until $t_{\max} = 140$. As for the complex Ginzburg-Landau equation, we perturb the solution of the limit cycle attractor. We again scale snapshots using a constant factor $p \in \{0.9, 1.1\}$ such that

$$\begin{pmatrix} V_k^{\mathrm{new}} \\ h_k^{\mathrm{new}} \end{pmatrix} = p \begin{pmatrix} V_k \\ h_k \end{pmatrix} \quad (25)$$

and integrate these perturbed snapshots forward in time for an interval of $t = 20$. We do this three times along the limit cycle to sample transients for training, and one extra time for testing. Finally, the sampled data is rescaled $V_k \rightarrow (V_k + 37)/30$ and $h_k \rightarrow (h_k - 0.42)/0.2$ such that both variables are approximately mean centered and are distributed over the same interval.

We employ diffusion maps with a kernel scale of $\epsilon = 4000$ based on earlier studies[9]. As in the previous sections, we scale the resulting diffusion eigenvectors onto the interval $[-1, 1]$. We fit the data on the rectangular grid shown in Fig. 5(b) using polynomials of maximal order two. The data is then interpolated on a mesh of 64 grid points in each direction.

The PDE model is represented by a neural network with three hidden layers of 64 neurons, each followed by a tanh activation function. The input at each point consists of the rescaled and interpolated $V_k$ and $h_k$ values, as well as their spatial derivatives in both $\phi_1$ and $\phi_2$ up to order three obtained using finite differences. Here, for simplicity, we do not use mixed derivatives. The model is optimized by minimizing the mean-squared error between its output and the temporal derivatives of $V_k$ and $h_k$ obtained through finite differences in time. For integration, we use the output of the neural network and step forward in time using forward Euler with $dt = 2 \cdot 10^{-3}$. Finally, we scale the resulting $V_k$ and $h_k$ back to their physical variables, as they are shown in Fig. 5.

For filtering, we keep 10 SVD modes, capturing more than 99.99% of the variance contained in the data.

## Data availability
The data generated in this study are provided in the Supplementary Information/Source Data file. All data can be reproduced using the code published under https://github.com/fkemeth/emergent_pdes. Source data are provided with this paper.

## Code availability
The source code to generate the reported data and to reproduce the results, as well as all figures, is available under https://github.com/fkemeth/emergent_pdes.

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

## Acknowledgements

This work was partially supported by the U.S. Army Research Office (through a MURI program), DARPA, and the U.S. Department of Energy (I.G.K., F.P.K., T.B., T.T.).

## Author contributions

I.G.K. conceived the research which was planned jointly with all the authors. F.P.K. performed a large part of the research, with contributions from T.B., T.T., F.D., S.J.M., and C.R.L. F.P.K. and I.G.K. initially wrote the manuscript, which was edited in final form with contributions from all the authors.

## Competing interests

The authors declare no competing interests.
