## [Peer Review File · Nature Communications]

Learning emergent partial differential equations in a learned emergent spaceREVIEWER COMMENTS

Reviewer #1 (Remarks to the Author):

This work develops a method to learn an effective PDE from data, even in cases where the data does not originate from a PDE. The first step is to learn effective spatial coordinates, which is accomplished via diffusion maps. In this step, the authors treat the time series of each agent (or mesh point) as a point in a high-dimensional space, and apply the diffusion maps method to the collection of time series, thereby grouping similar time series together in the diffusion map coordinates. The diffusion map coordinates thus smoothly parameterize the time series. In the second step, the authors learn a PDE for the data, where the right-hand-side is a function of derivatives with respect to the diffusion map coordinates, and well-documented methods are used to learn the right-hand-side. The authors show that modeling transient solutions in this way results in good short- and long-time predictions.

There are some interesting ideas here. The authors show that they can learn a PDE from data that originates from a PDE, as well as from data that does not originate from a PDE with their unique approach to finding "emergent" spatial variables. On the other hand, the application examples are relatively simple, the present work is largely a logical extension of prior studies from the authors, and the presentation is often unclear. Ultimately, I do not view the work as a substantial enough advance to warrant publication in Nat Comm.

Comments:

1. This paper combines previous work to learn effective spatial coordinates (ref. 9) with well-known work to learn the right-hand-side of a system of ODEs, and is essentially a logical extension of ref. 9.
2. Solving a PDE is typically expensive; can the authors give an example where solving a PDE is less expensive than the original system of oscillators? How do the costs compare for the Stuart-Landau example? Moreover, a common way to solve PDEs is to project onto some finite basis. Why not go directly to this finite representation instead of going through a PDE? What are the benefits of going through a PDE?
3. Related to the previous point, the dynamics in the two examples are simply limit cycles. For such simple dynamics, I think it would be better to go straight to a very low-dimensional representation of the dynamics instead of going through a PDE. Does the method work for more complicated dynamics? An example with more complicated dynamics would be beneficial.
4. The authors mention interpretability as a benefit or possible feature of learning a PDE, but I find this to be an unlikely feature when the effective spatial coordinates are learned from data. In the two examples in the paper, the authors were only able to interpret the diffusion map coordinate because they already knew the answer or had a very small set of possible answers.
5. More information is needed about the parameters used in the two examples so that readers could recreate the data if they wanted to. Also, it would be useful to explain why the authors chose the parameters values (and initial and boundary conditions) that they did. Was this simply to get limit cycle behaviour? I also thought that the Methods section was a bit opaque, and think it needs to be expanded and made more explicit.
6. The way boundary conditions are dealt with seems like a rather large weakness to me.

This is especially the case for the Stuart-Landau example. From fig. 3d, we can infer that nearly all of the oscillators lie in a range of ϕ_1 values in the boundary corridor illustrated in fig. 3b. The PDE is only learned in the interior region, which ostensibly contains very few oscillators. So, it seems that a PDE is being learned in order to replace very few oscillators, which would be rather wasteful. Is there a way to learn effective boundary conditions instead of the current way of dealing with them?

7. Are the values of the diffusion map coordinate always between -1 and 1, or have they been rescaled and shifted? Of the many potential diffusion map coordinates, how do the authors know which one to pick? How do the authors know how many effective spatial coordinates are needed, and how do they pick the appropriate diffusion map coordinates? Also, what do the authors mean by "independent" (line 231)? Will mixed derivatives show up, and why are they not considered in eq. 7?

8. What advantages do diffusion maps offer for finding the "emergent" spatial variable? It is commented that if it is helpful this can be thought of similarly to keeping the leading component of PCA. Would PCA work? Would other dimension reduction methods work like tSNE, LLE, Isomaps, undercomplete autoencoder, etc.?

9. It would be helpful to more explicitly explain the fixed boundary conditions. It sounds like the boundary conditions are just fixed to some value and this effects the interior points through the derivative calculations. What happens when the boundary conditions are not enforced, do solutions blow up? This constraint seems unnecessary with a good approximation of f .

10. The examples explored are no more complicated than periodic orbits. Would this method work for chaotic datasets? The authors mention swarms; agent-based models for these are simple to code up. How about looking at this case?

11. Why is the projection onto the leading modes from SVD important? Does this smooth out higher frequency behavior that causes solutions to blow up with the neural network model?

12. The authors show that they are able to learn an effective local PDE for globally coupled oscillators. Although this seems contradictory, certain types of local PDEs do exhibit global solutions (e.g., parabolic PDEs like the heat equation). Something to this effect should be added to the manuscript.

13. How necessary is it to provide derivatives to the neural network? It would be useful to compare the performance when no derivatives are supplied, then the first, and the second. No derivatives is also useful because it shows the effect of learning the "emergent" spatial variables.

14. In addition to showing the distance between the transient and the true attractor (fig 2f,3c,5d) it would be helpful to see the distance between the true transient and the model to judge how well short-time tracking performs.

15. For the two "emergent" spatial variables why are derivatives with respect to both variables not needed?

16. There should be another figure like 5a showing λ , or it could be made into a 3D figure to show λ .

17. The description of the grid for the two "emergent" spatial variables is very unclear. Why

is this region selected? How can interpolation be done in the bottom right corner where no data points exist? Why are a majority of the omega values dropped?

18. I find the abstract to be too indirect. It would be nice if it were made more clear that a neural network is used for the model and diffusion maps are used to reorganize the data. It seems like the key points ought to be: 1. systems of coupled oscillators have no obvious spatial relation 2. "emergent" spatial variables are discovered via diffusion maps 3. in this emergent space partial differential equations are approximated with neural networks.

19. Some typos: "keepinging" on line 140, "knowing knowing" on line 145, " $n_{\text{train}}=1$ " looks like it should be " $n_{\text{test}}=1$ " on line 426

Reviewer #2 (Remarks to the Author):

The authors present a methodology for learning/describing the evolution of a discrete system through neural networks that use as inputs discrete approximations of PDE kernels. The paper contains several original ideas by the authors (many pioneered and presented by the authors in previous studies) on how to accelerate fine scale simulations by discovering an effective coarse grained PDE. The new "twist", as the authors clearly explain, is the identification of a previously unknown independent variable.

I find the presentation of these ideas as a very interesting read.

However right from the start it is clear that they do not learn a PDE but rather a neural net representation of discrete approximations of PDEs. Even more they "help" the NN learn the terms of the PDE that is clearly linked to the respective discrete approximation. So there is limited hope that this methodology may work when such information is not available (as it would not be available on most agent based simulations). What if the "wrong" derivatives had been fed to the NN ? Would the method still work? My estimate is that it would not as after all the NNs can only interpolate and not discover dynamics.

Furthermore, I am disappointed to see that despite the authors claiming important advances to fields ranging from chemistry to quantum mechanics and fluid flows the results involve a 1D example with rather smooth solutions. The authors claim "dramatic" savings in computation "if" this idea is successful, but I do not see any of these claims being supported by the 1D example presented in this paper.

Despite my concerns with the approach I believe that the paper will gain importance (and I hope that it would prove me wrong) by showcasing a problem that is in one of the following categories:

1. 2D or 3D in space and with complex boundary conditions
2. has complex and non-smooth/decaying solutions (see figure 3c on what is the actual and learned)
3. is truly agent based and a discretization of a PDE that is in the end "rediscovered"

Perhaps ideas deserve as much attention as proof that they work. Hence I would not object to the publication of this paper if it is decided by the journal.

In summary, I am not convinced that this idea would work to anything beyond smooth solutions due to the issues I outline above. If this is the case what is the value of this new approach over existing methodologies (machine learning and/or other coarse graining procedures).

At the same time, I will be happy to stand corrected and would be glad to see a a revised

version of the paper that addresses my concerns and most importantly that it tackles problems in one of the the 3 categories described above.

Reviewer #3 (Remarks to the Author):

I appreciate what the authors are trying to do here: take the dynamics from a set of coupled agents whose equations of motion are **_unknown_**, and then use this to derive a data driven PDE to describe the resulting dynamics. For such an approach to be useful, the dynamical laws that are so derived must (i) apply accurately within the domain upon which the model was trained (eg initial conditions), and ideally (ii) also generalize to unseen parameters.

I'm concerned that the choice of model problems (+ parameter ranges) that are chosen in this manuscript are too simplistic, and for that reason the idea doesn't deliver on its promise. The first example where x is the coordinate is especially simplistic -- the original PDE already exists in these variables. The second one (Stuart) is getting closer to the complexity where something interesting could be found.

The approach that the authors take is sensible and interesting: they use a data driven method (kernel based PCA) to derive modes from data and then try to use the leading components as a coordinate system to derive the equations. I like this approach very much. There is however a straw man: we could take the modes, which eg in the 2nd Stuart example (Fig 3d) the authors demonstrate a 1-1 mapping between ϕ_1 and ω_i . Given this mapping could we not directly derive a PDE for $W(\phi_1, t)$ by simply plugging it back into equation 4? The coupling term $(K/N \sum \dots)$ induces derivatives in the ϕ_1 coordinates. The resulting PDE is on the surface close to that of Eqn 5 except that instead of using neural networks it can be derived directly from the equations. It also has the advantage that the parameter variation in this equation is clear -- so it is more generally applicable. Presumably this equation will give rise to the bifurcations explored in Fig 4.

Other comments:

--The fact that all the analysis of the paper is close to a bifurcation point is worrisome. Would it make sense of a system where the dynamical behavior was much more complicated--so that perhaps 2 principle components or more are actually needed? For example a simple challenge is to redo the PDE example in eqn 1 for either the KS equation or for Lorenz 96 model, which shows truly chaotic behavior.

--One major issue with the simplicity of the dynamical systems explored here is that it is unclear how much of what is being demonstrated is memorization. Can you demonstrate that the models work on out of distribution test examples? Choosing different random initial conditions of these models in these regimes does not accomplish this--given the simplicity of the dynamics what happens is highly constrained/has low entropy and hence agreement=memorization.

--Agent based models with complex dynamics abound, much more complicated than those outlined here. For example one area the authors might wish to explore are covid -agent based models (eg VaTech, Northeastern,...). The rules for the agents can be quite complicated, and there is an obvious low order dynamical system at the core. Could the present approach actually be useful in that regard--deriving effective dynamics that are predictive for out of distribution examples?

The "dream" behind this work is quite beautiful--but I'm afraid at this point the results could be made much more convincing by choosing better examples.

Dear Editor, Dear Reviewers

We have taken the reviewers comments very seriously, and -beyond the thoughtful edits, corrections, clarifications... they suggested- we went on and performed -and included in the paper- a different and significantly more complex example: a large network of heterogeneous neurons (arising in modeling the pre-Boetzing complex); neurons that are heterogeneous both kinetically and structurally (they have a Chung-Lu network type connectivity). This is a problem for which there is no a-priori physical space that we artificially scrambled to then recover it; yet we knew we could find a two-dimensional emergent space, and now we went on and learned an effective emergent PDE in this space. We are particularly happy to report that we can also establish a (nontrivial) interpretation for these emergent space coordinates (they can be mapped to features of the neuron heterogeneity).

We very much hope that the careful response to the helpful and constructive comments, augmented by this (in our opinion seriously nontrivial) example will make the paper worthy of publication in the Journal.

OUR DETAILED RESPONSES FOLLOW:

Reviewer #1 (Remarks to the Author):

This work develops a method to learn an effective PDE from data, even in cases where the data does not originate from a PDE. The first step is to learn effective spatial coordinates, which is accomplished via diffusion maps. In this step, the authors treat the time series of each agent (or mesh point) as a point in a high-dimensional space, and apply the diffusion maps method to the collection of time series, thereby grouping similar time series together in the diffusion map coordinates. The diffusion map coordinates thus smoothly parameterize the time series. In the second step, the authors learn a PDE for the data, where the right-hand-side is a function of derivatives with respect to the diffusion map coordinates, and well-documented methods are used to learn the right-hand-side. The authors show that modeling transient solutions in this way results in good short- and long-time predictions.

There are some interesting ideas here. The authors show that they can learn a PDE from data that originates from a PDE, as well as from data that does not originate from a PDE with their unique approach to finding "emergent" spatial variables. On the other hand, the application examples are relatively simple, the present work is largely a logical extension of prior studies from the authors, and the presentation is often unclear. Ultimately, I do not view the work as a substantial enough advance to warrant publication in Nat Comm.

We thank reviewer for the insightful and very constructive remarks on our article (and hope we may change your view about publication). We made corresponding changes in the article and address each the point raised in more detail below.

Comments:

1. This paper combines previous work to learn effective spatial coordinates (ref. 9) with well-known work to learn the right-hand-side of a system of ODEs, and is essentially a logical extension of ref. 9.

Yes, the work presented here is a logical (and, we would like to think, important) extension of our work on constructing emergent spaces. The present work (a) contains a demonstration that learning

a PDE in this space is indeed possible, and (b) highlights important issues that arise in attempting this task: some numerical, such as the SVD regularization to prevent instabilities, and some conceptual/mathematical/numerical, such as the required boundary conditions for the learned equation, and the question of its well-posedness more generally. We are convinced that realizing the ideas described in our earlier work, and discussing the issues that arise in doing so, are worth sharing, and will also provide a useful step towards automated data-driven system identification.

2. Solving a PDE is typically expensive; can the authors give an example where solving a PDE is less expensive than the original system of oscillators? How do the costs compare for the Stuart-Landau example? Moreover, a common way to solve PDEs is to project onto some finite basis. Why not go directly to this finite representation instead of going through a PDE? What are the benefits of going through a PDE?

The Stuart-Landau ensemble is a particularly simple model where the network topology is all-to-all coupling. In general, and in the non-all-to-all coupled neuron example added in this revised version, the topology is much more complicated. In this latter added example, the degree differs from neuron to neuron. Performing bifurcation analysis on such a problem with the original system becomes very cumbersome even for a moderate number of neurons, since one has to (a) figure out which neuron is coupled to which and (b) parse all the neuron states in the neighborhood of each neuron.

In contrast, having a PDE description in emergent coordinates allows the use of well-established discretization techniques and bifurcation tools for PDEs. In addition, the PDE can now be solved with an optimized number of spatial grid points. This number can be much less than the number of oscillators in the original system (for example in the Stuart-Landau example in the paper, where we sample data from a system of 512 oscillators but integrate the PDE on a domain with 128 grid points, see Methods). Using the original ensemble representation essentially is doing an unprincipled finite-difference or finite-element discretization of the problem, and so (if we are to believe that there's an underlying continuous system) the least efficient way to simulate. Explicitly modeling the dynamics with a PDE would let a practitioner develop more efficient discretizations.

However, we agree with the reviewer that in some cases, and in particular when the agent ensemble is small to begin with, and the number of space dimensions is large, solving the PDE might be computationally more expensive. We therefore added the following sentence in the introduction: "This is in particular the case when the agent ensemble is large but the set of agents can be parametrized with only a few emergent parameters."

In a more general context, it is also of scientific interest if a PDE exists at all for a given oscillator system - and if it does, what dimension, number of necessary derivatives, etc. are necessary for it to close.

3. Related to the previous point, the dynamics in the two examples are simply limit cycles. For such simple dynamics, I think it would be better to go straight to a very low-dimensional representation of the dynamics instead of going through a PDE. Does the method work for more complicated dynamics? An example with more complicated dynamics would be beneficial.

We added another example to the revised version, where we use a model describing the synchronization phenomena in the preBötzinger complex in the brain. This model consists of randomly coupled Hodgkin-Huxley-like neurons, which, as we discover, can be embedded in a two-dimensional emergent space. The model is more complicated in that it has a structural heterogeneity (random coupling) and intrinsic heterogeneity (heterogeneous parameter that differs from neuron to neuron).

To illustrate our approach, and to verify that the learned PDE accurately captures the long-term dynamics, we chose examples with non-chaotic dynamics. We are confident about the ability to extend the approach to low-dimensional chaotic dynamics (we are much less confident about turbulent systems, where the attractor can become infinite-dimensional).

Of course, using chaotic examples would render the comparison between the learned model and the true dynamics more difficult, because one would have to check that the predicted trajectories diverge with the right Lyapunov exponent and that, at the same time, the attractors of both systems reasonably coincide.

4. The authors mention interpretability as a benefit or possible feature of learning a PDE, but I find this to be an unlikely feature when the effective spatial coordinates are learned from data. In the two examples in the paper, the authors were only able to interpret the diffusion map coordinate because they already knew the answer or had a very small set of possible answers.

The reviewer is right that one must have some physical quantities in mind for which to test if they correspond to the embedding coordinates obtained using manifold learning. We therefore stated that some “domain knowledge” in the form of possible physical quantities that may explain the emergent coordinates has to be available.

In our first example the density sounds indeed like an “obvious suspect” – but in the new example we included now, the physical interpretation of the emergent space coordinates is not so obvious. Be that as it may, when one has such physical quantities, automated algorithms exist (e.g. from Marina Meila, Ref. 48 in the revised version, or from Sunday et al, Ref. 46 in the revised version) that test the explainability of the emergent coordinates through these quantities.

Our approach then frees the experimentalist from having to choose a minimal and predictive number of physical quantities to model the dynamics, they simply need to test which subset(s) can be mapped invertibly to the emergent coordinates (which already are good, predictive coordinates by construction).

5. More information is needed about the parameters used in the two examples so that readers could recreate the data if they wanted to. Also, it would be useful to explain why the authors chose the parameters values (and initial and boundary conditions) that they did. Was this simply to get limit cycle behaviour? I also thought that the Methods section was a bit opaque, and think it needs to be expanded and made more explicit.

Thank you for pointing this out. We added the following sentences in the Methods section to clarify the issues:

For the CGLE example:

“It is worth noting that there is a slight left-right asymmetry in the solution shown in Fig. 3(a). Due to the symmetry of the spatial domain, there exist two stable solutions for this set of parameters; one has a slightly larger amplitude for large x ; the other, reflected version, has a larger amplitude for small x . Choosing the initial condition defined above leads to a convergence to the same solution at every run. However, all initial conditions will eventually come down to a periodic, symmetry related, solution.”

For the Stuart-Landau example:

“We integrate Eq. 1 using an implicit Adams method with the initial conditions of the oscillators uniformly distributed in the unit square in the complex plane. The intrinsic frequencies are thereby linearly spaced in the interval $[-1.5, 1.9]$, and the coupling constant is taken as $K=1.2$.”

6. The way boundary conditions are dealt with seems like a rather large weakness to me. This is especially the case for the Stuart-Landau example. From fig. 3d, we can infer that nearly all of the

oscillators lie in a range of ϕ_1 values in the boundary corridor illustrated in fig. 3b. The PDE is only learned in the interior region, which ostensibly contains very few oscillators. So, it seems that a PDE is being learned in order to replace very few oscillators, which would be rather wasteful. Is there a way to learn effective boundary conditions instead of the current way of dealing with them?

Yes, there are multiple ways to learn effective boundary conditions, for example through direct supervised learning in case the data is available, or via more involved Green's function approaches (<https://doi.org/10.1098/rspa.2021.0229>). Special care has to be taken to ensure that there are “enough” boundary conditions to make the problem well defined – the interplay between machine learning and problem well-posedness is only now starting to arise as a research direction.

Here, we chose to provide small corridors at the boundary of the domain, that are wide enough to allow the computation of all required derivatives (and thus make the problem well-posed).

If one knows the functional form of the boundary conditions, and what is missing is only a couple of parameter values, this can also be easily accomplished as part of the training – but working in emergent space exacerbates the difficulty.

To keep our presentation simple, we chose not to learn the boundary conditions, but to just provide the corridors of true data. Identifying well-posedness through machine learning is a very important (and to our knowledge) unexplored research direction. Our simple approach circumvents the difficulties – we are working separately on this direction, but for simpler, “cleaner” problems where we do not have the additional twist of having to construct an emergent space first.

The reviewer is right that in the example of the Stuart-Landau oscillators, many oscillators are located at the edges of the emergent space domain, and one cannot expect large computational savings in the interior for the ensemble size of 512 oscillators considered here, see Methods.

However, computational savings can be expected when investigating larger systems.

In addition, the condensation of oscillators at the emergent space boundaries does not hold for other coupled oscillator systems in general, see for example the preBötzinger system added to the revised version.

7. Are the values of the diffusion map coordinate always between -1 and 1, or have they been rescaled and shifted? Of the many potential diffusion map coordinates, how do the authors know which one to pick? How do the authors know how many effective spatial coordinates are needed, and how do they pick the appropriate diffusion map coordinates? Also, what do the authors mean by “independent” (line 231)? Will mixed derivatives show up, and why are they not considered in eq. 7?

We thank the reviewer especially for this important remark. We added the following sentences when describing diffusion maps in the Methods section:

Note that the eigenvectors of the diffusion matrix correspond to the eigenfunctions of the Laplace Beltrami operator on the data manifold. As such, eigenvectors that can be written as functions of other eigenvectors with larger eigenvalue appear in the eigendecomposition of the diffusion matrix. An important task when using diffusion maps is to extract the independent eigenvectors that parametrize new directions in the data.

A prominent tool for this task was developed in Ref. [Dsilva et al., 2018] and is based on performing local linear regression on the set eigenvectors. Here, we perform visual inspection of the first ten eigendirections to investigate which eigenvectors are harmonics and which eigenvectors represent new directions in the data.

These independent diffusion modes are then scaled to the interval [-1, 1] for better comparison.

We did not use, for simplicity, mixed derivatives in space. There are recent works investigating how many derivatives are necessary for learning the right-hand side of a PDE, see for example the following sentence and reference in our discussion:

“An important question in deciding which PDE model to learn, is how many “emergent spatial” derivatives one has to include in the PDE right hand side. In other words, how can one decide when $\partial W/\partial t$ is well approximated by W and its derivatives with respect to ϕ_1 ? For Gaussian process regression, recent work using Automatic Relevance Determination helps tackle this problem [41].”

8. What advantages do diffusion maps offer for finding the "emergent" spatial variable? It is commented that if it is helpful this can be thought of similarly to keeping the leading component of PCA. Would PCA work? Would other dimension reduction methods work like tSNE, LLE, Isomaps, undercomplete autoencoder, etc.?

In principle, any manifold learning technique would work for this task. Here, we chose diffusion maps. One reason is that it is nonlinear as opposed to PCA, and so in principle more parsimonious. This is advantageous in the Stuart-Landau example, where the emergent space is actually an open ring (see figure 4c). In general, non-linear techniques are advantageous for data manifolds with high curvature, where linear techniques would result in far too many embedding coordinates. When the goal is to learn PDE in those coordinates, parsimony is even more important, which is why we chose to demonstrate the approach with such a non-linear technique.

9. It would be helpful to more explicitly explain the fixed boundary conditions. It sounds like the boundary conditions are just fixed to some value and this effects the interior points through the derivative calculations. What happens when the boundary conditions are not enforced, do solutions blow up? This constraint seems unnecessary with a good approximation of f .

For the PDE initial value problem to be well posed, sufficient initial/ boundary conditions must be provided. But knowing what is “just sufficient” to guarantee well-posedness for a problem where the PDE operator is “hidden” in a neural network is highly nontrivial – we believe that it is an important new direction in ML research, and we have in the past spent some effort in “deciding the nature” of the unavailable equation (SIAM Review 49(3) pp.469-487, 2007).

Here, instead of trying to find minimal boundary conditions, we provide small corridors of true data on the boundaries of the domain, which are thick enough to calculate all the required derivatives in the interior. This should provide enough information, as long as the data in the corridors are physical and compatible (that is, they come from a simulation). However, the observations along the boundary corridors have to be available for us to provide them to the solver.

In principle, and as also discussed in point 6 above, one could also try to learn or fit sufficient boundary conditions – a whole new endeavor by itself. We refrained from doing this here and circumvented the question of “how many boundary conditions are enough” - but we believe that this is an exciting topic for future research. We do currently work on ill-posedness of learned PDE models: what happens when we do not provide enough boundary conditions to the model, and how that can be observed or established in a data driven way. We expect to be able to report on it within this calendar year.

10. The examples explored are no more complicated than periodic orbits. Would this method work for chaotic datasets? The authors mention swarms; agent-based models for these are simple to code up. How about looking at this case?

To illustrate our approach, and to verify that the learned PDE accurately captures the long-term dynamics, we chose examples with non-chaotic dynamics. Using chaotic examples would render the comparison between the learned model and the true dynamics even more complicated, because one would have to check that the predicted trajectories diverge with the right Lyapunov exponent and that, at the same time, the attractors of both systems coincide.

We did do some additional work towards increased complexity: As we also stated in response to point 3 above, we added a biologically motivated system of coupled Hodgkin-Huxley type neurons, used to model the dynamics in parts of the brain, to illustrate that our approach is invariant to the detailed model.

11. Why is the projection onto the leading modes from SVD important? Does this smooth out higher frequency behavior that causes solutions to blow up with the neural network model?

Yes. Having access to training data from only a finite number of dimensions leaves the model ignorant to the dynamics in the remaining dimensions (where the dynamics are most probably attracting, but we do not have enough data to learn this). During prediction, this can lead to instabilities in these dimensions (typically high frequency ones), which grow during integration. To prevent such instabilities, which originate from limited access to training data, we filter the output from time to time during prediction. However, other approaches might work as well, which we want to explore in the future. One example is to use hyperviscosity, damping high frequency modes during prediction. We therefore added the following sentence in the discussion:

“Other approaches may be employed as well, such as hyperviscosity in the learned PDE model [38,39,40], effectively damping higher frequency components.”

12. The authors show that they are able to learn an effective local PDE for globally coupled oscillators. Although this seems contradictory, certain types of local PDEs do exhibit global solutions (e.g., parabolic PDEs like the heat equation). Something to this effect should be added to the manuscript.

This is a very good point! This relates to the problem of “when does a PDE description of the dynamics exist”. Here, we argue in the introduction and discussion that when the dynamics is smooth and (ultimately) low dimensional, we can take advantage of the Takens embedding theorem and use generic observations (such as spatial derivatives) as observables to describe the dynamics.

We added a paragraph in the discussion accordingly:

“In our example, the oscillators are globally coupled, while we find a local PDE (without integral terms) that successfully describes their behavior. This apparent disconnect of local PDE description versus global coupling can be explained through infinite propagation speed of information for certain parabolic PDE, such as the heat equation. Modeling globally coupled oscillators with a PDE that only allows finite propagation speed, such as the wave equation, would not lead to the correct behavior. In our case, the network automatically learned that infinite propagation speed is necessary, and we are still investigating how such qualitative behavior can be learned more effectively.”

The question, which is still under current investigation, is when the right-hand side of the PDE is a function (functional) of the observables we have chosen to retain.

13. How necessary is it to provide derivatives to the neural network? It would be useful to compare the performance when no derivatives are supplied, then the first, and the second. No derivatives is also useful because it shows the effect of learning the "emergent" spatial variables.

Note that here we learn a translationally invariant PDE. This means the dynamics does not depend on the emergent space coordinates, but only on the variables and derivatives of them with respect to the emergent space coordinates.

Without derivatives as input (without information from even a few of one oscillator's neighbors in behavior) one would effectively learn a decoupled ODE which would have to be valid at each point in space and time.

One can furthermore check which derivatives are important for the right-hand side of a PDE model using methods such as automated relevance determination, see for example Ref. [41] in the revised version.

14. In addition to showing the distance between the transient and the true attractor (fig 2f,3c,5d) it would be helpful to see the distance between the true transient and the model to judge how well short-time tracking performs.

Below is a plot of the learned and true real parts of dW/dt for the Stuart-Landau system. Due to space constraints, we do not show this figure in the article. However, we do think that showing the distance plots, Figs. 2f and 3c, indicate that the short-time performance is good as well.

15. For the two "emergent" spatial variables why are derivatives with respect to both variables not needed?

We did not use, only for simplicity, mixed derivatives in space. However, as we also discuss in the article, any $2n+1$ observations of the field, with n being the dimension of the attractor, are sufficient as input to the PDE model. However, we agree that one could also use mixed derivatives (for example instead of higher order separate derivatives).

16. There should be another figure like 5a showing lambda, or it could be made into a 3D figure to show lambda.

We agree. In the revised version, we replaced the 2d Stuart-Landau example with the preBötzinger model of coupled neurons. There, we colored the diffusion maps coordinates with the intrinsic heterogeneous parameter (as figure 5a in the first version). We do not show the embedding colored with the other heterogeneous parameter, since it has been published earlier (see reference in the text)

17. The description of the grid for the two "emergent" spatial variables is very unclear. Why is this region selected? How can interpolation be done in the bottom right corner where no data points exist? Why are a majority of the omega values dropped?

In principle, no interpolation/extrapolation is needed when learning the (discretized) PDE if one resorts to using finite differences on an irregular grid.

During predictions, however, a domain and boundary conditions at the edges of the domain have to be specified for the problem to be well-defined. By interpolating (and also extrapolating on the edges of) a rectangular grid, these tasks become easier, and as we hope, the idea more illustrative.

18. I find the abstract to be too indirect. It would be nice if it were made more clear that a neural network is used for the model and diffusion maps are used to reorganize the data. It seems like the key points ought to be: 1. systems of coupled oscillators have no obvious spatial relation 2. "emergent" spatial variables are discovered via diffusion maps 3. in this emergent space partial differential equations are approximated with neural networks.

Thank you for pointing this out! We changed the abstract to:

We propose an approach to learn effective evolution equations for large systems of interacting agents. This is demonstrated on two examples, a well-studied system of coupled normal form oscillators and a biologically motivated example of coupled Hodgkin-Huxley-like neurons. For such types of systems, there is no obvious space coordinate in which to learn effective evolution laws in the form of partial differential equations.

In our approach, we accomplish this by learning embedding coordinates from the time series data of the system using manifold learning as a first step.

In these "emergent" coordinates, we then show how one can learn effective partial differential equations, using neural networks, that do not only reproduce the dynamics of the oscillator ensemble, but also capture the collective bifurcations when system parameters vary.

The proposed approach thus integrates the automatic, data-driven extraction of emergent space coordinates parametrizing the agent dynamics, with machine-learning assisted identification of an "emergent PDE" description of the dynamics in this parametrization.

19. Some typos: "keepinging" on line 140, "knowing knowing" on line 145, " $n_{\text{train}}=1$ " looks like it should be " $n_{\text{test}}=1$ " on line 426

We fixed the typos, thank you for pointing them out!

Reviewer #2 (Remarks to the Author):

The authors present a methodology for learning/describing the evolution of a discrete system through neural networks that use as inputs discrete approximations of PDE kernels. The paper contains several original ideas by the authors (many pioneered and presented by the authors in previous studies) on how to accelerate fine scale simulations by discovering an effective coarse

grained PDE. The new "twist", as the authors clearly explain, is the identification of a previously unknown independent variable.

I find the presentation of these ideas as a very interesting read.

However right from the start it is clear that they do not learn a PDE but rather a neural net representation of discrete approximations of PDEs. Even more they "help" the NN learn the terms of the PDE that is clearly linked to the respective discrete approximation.

So there is limited hope that this methodology may work when such information is not available (as it would not be available on most agent based simulations). What if the "wrong" derivatives had been fed to the NN ? Would the method still work? My estimate is that it would not as after all the NNs can only interpolate and not discover dynamics.

Furthermore, I am disappointed to see that despite the authors claiming important advances to fields ranging from chemistry to quantum mechanics and fluid flows the results involve a 1D example with rather smooth solutions. The authors claim "dramatic" savings in computation "if" this idea is successful, but I do not see any of these claims being supported by the 1D example presented in this paper.

Despite my concerns with the approach I believe that the paper will gain importance (and I hope that it would prove me wrong) by showcasing a problem that is in one of the following categories:

1. 2D or 3D in space and with complex boundary conditions
2. has complex and non-smooth/decaying solutions (see figure 3c on what is the actual and learned)
3. is truly agent based and a discretization of a PDE that is in the end "rediscovered"

Perhaps ideas deserve as much attention as proof that they work. Hence I would not object to the publication of this paper if it is decided by the journal.

In summary, I am not convinced that this idea would work to anything beyond smooth solutions due to the issues I outline above. If this is the case what is the value of this new approach over existing methodologies (machine learning and/or other coarse graining procedures).

At the same time, I will be happy to stand corrected and would be glad to see a revised version of the paper that addresses my concerns and most importantly that it tackles problems in one of the the 3 categories described above.

We thank the reviewer for the insightful and constructive remarks on our article. We added another (we hope you will agree, nontrivial!) example to the revised version where we use an agent-based model describing the synchronization phenomena in the preBöttinger complex in the brain. This model consists of randomly coupled Hodgkin-Huxley like neurons, which can be embedded in a two-dimensional emergent space. The model is insofar more complicated in that it has a structural heterogeneity (random coupling) and intrinsic heterogeneity (heterogeneous parameter that differs from neuron to neuron). However, the system as a whole is, for this set of parameters, also periodic. Also (admittedly with the help of discretization) we learn an approximation of the law (the right-hand-side operator) of the PDE; please note that our novelty is NOT the learning of the PDE; that was novel in 1998 and very fashionable again now in machine learning; what we feel is our novelty is in fusing the data mining with the PDE learning in a novel (we hope) way.

To illustrate our approach, and to verify that the learned (approximate) PDE accurately captures the long-term dynamics, we chose examples with non-chaotic dynamics. Using chaotic examples would render the comparison between the learned model and the true dynamics more difficult, because one would have to check that the predicted trajectories diverge with the right Lyapunov exponents and that, at the same time, the attractors of both systems coincide.

The Stuart-Landau ensemble is a particularly simple model where the network topology is all-to-all coupling. In general, and also in the coupled neuron example added in this revised version, the topology is much more complicated. Here, already the degree differs from neuron to neuron. Performing bifurcation analysis on such a problem becomes very cumbersome even for a moderate number of neurons, since one has to (a) figure out which neuron is coupled to whom and (b) parse all the neuron states in the neighborhood of each neuron.

In contrast, having a PDE description allows the usage of well-established bifurcation tools for PDEs.

In addition, the PDE can now be solved with a minimal number of spatial grid points. In particular, this can be much less than the number of oscillators in the original system (as for example in Stuart-Landau example in the paper, where we sample data from a system of 512 oscillators, but integrate the PDE on a domain with 128 grid points, see Methods).

However, we agree with the reviewer that in some cases, and in particular when the agent ensemble is small to begin with, and the number of space dimensions is large, solving the PDE might not be computationally cheaper.

We therefore added the sentence in the introduction after our claims:

“This is in particular the case when the agent ensemble is large but the set of agents can be parametrized with only a few emergent parameters.”

As is shown in more detail in Ref. 41 in the revised article, there is a multitude of derivatives that serve as sufficient input to the PDE model. That is, as long as we provide enough derivatives to the model, we can predict the dynamics. “Enough” hereby depends on the dynamics, and the actual number can be calculated using methods such as automated relevance determination.

Reviewer #3 (Remarks to the Author):

I appreciate what the authors are trying to do here: take the dynamics from a set of coupled agents whose equations of motion are $_unknown_$, and then use this to derive a data driven PDE to describe the resulting dynamics. For such an approach to be useful, the dynamical laws that are so derived must (i) apply accurately within the domain upon which the model was trained (eg initial conditions), and ideally (ii) also generalize to unseen parameters.

I'm concerned that the choice of model problems (+ parameter ranges) that are chosen in this manuscript are too simplistic, and for that reason the idea doesn't deliver on its promise. The first example where x is the coordinate is especially simplistic -- the original PDE already exists in these variables. The second one (Stuart) is getting closer to the complexity where something interesting could be found.

We thank the reviewer for their insightful and constructive remarks on our article. We made corresponding changes in the article, and we address each the point raised in more detail below.

The approach that the authors take is sensible and interesting: they use a data driven method (kernel based PCA) to derive modes from data and then try to use the leading components as a coordinate system to derive the equations. I like this approach very much. There is however a straw man: we could take the modes, which eg in the 2nd Stuart example (Fig 3d) the authors demonstrate a 1-1 mapping between ϕ_1 and ω_i . Given this mapping could we not directly derive a PDE for $W(\phi_1, t)$ by simply plugging it back into equation 4? The coupling term $(K/N \sum \dots)$ induces derivatives in the ϕ_1 coordinates. The resulting PDE is on the surface close to that of

Eqn 5 except that instead of using neural networks it can be derived directly from the equations. It also has the advantage that the parameter variation in this equation is clear -- so it is more generally applicable. Presumably this equation will give rise to the bifurcations explored in Fig 4.

This is a very interesting thought. Ignoring diffusion maps, and given we would know the intrinsic frequencies of the oscillators, it is true that say

$$(W_{k+1}-W_k)/\Delta \approx \partial W/\partial \omega$$

where Δ is the spacing between the ω 's, and such a term occurs in the sum in (4) (up to some scaling). But there are many more terms in the sum in (4) and we do not see how they can all be rearranged into finite difference approximations to the various derivatives with respect to ω . Trying to do something similar for $W(\phi_1, t)$ would be even harder due to the nonuniform spacing of ϕ_1 values.

However, we think the referee's comment is very interesting, and reminds us of a similar approach for nonlocally coupled phase oscillators, where a PDE description could be derived (e.g. the paper <https://arxiv.org/pdf/1401.3089.pdf>)

Our second example would be even more "nontransparent" to match to finite differences -- and if we have the observations, but not the generating large set of ODEs, we know how many spatial variables we need, but not how to "plug them back into" the original model.

Other comments:

--The fact that all the analysis of the paper is close to a bifurcation point is worrisome. Would it make sense of a system where the dynamical behavior was much more complicated--so that perhaps 2 principle components or more are actually needed? For example a simple challenge is to redo the PDE example in eqn 1 for either the KS equation or for Lorenz 96 model, which shows truly chaotic behavior.

We added another example to the revised version, where we use a model describing the synchronization phenomena in the preBötzing complex in the brain. This model consists of randomly coupled Hodgkin-Huxley like neurons, which can be embedded in a two-dimensional emergent space. The model is more complicated insofar as it has a structural heterogeneity (random coupling) and intrinsic heterogeneity (heterogeneous parameter that differs from neuron to neuron). However, the system as a whole is, for this set of parameters, also periodic.

To illustrate our approach, and to verify that the learned PDE accurately captures the long-term dynamics, we chose examples with non-chaotic dynamics. Using chaotic examples would render the comparison between the learned model and the true dynamics more difficult, because one would have to check that the predicted trajectories diverge with the right Lyapunov exponents and that, at the same time, the attractors of both systems coincide. It is clear that this is worth trying for additional problems, but we did not go that far in this paper, as it would have added the above complications which are less related to learning PDE in an emergent space, and more related to the comparison of chaotic systems.

--One major issue with the simplicity of the dynamical systems explored here is that it is unclear how much of what is being demonstrated is memorization. Can you demonstrate that the models work on out of distribution test examples? Choosing different random initial conditions of these models in these regimes does not accomplish this--given the simplicity of the dynamics what happens is highly constrained/has low entropy and hence agreement=memorization.

When we sample initial conditions of the true system and the corresponding transients under true dynamics, we always take aside some of these transients as test data. To create the distance plots

and integration plots shown in the article, we always use one of the test initial conditions (which the model has not seen before) and integrate forward in time. This is already a simple test of whether the model generalizes. We furthermore carefully tested if the model is invariant under the grid point resolution (i.e. if the finite difference estimation of the partial derivatives are good) and, in figure 4, we show integration results of the learned model for time intervals (10000 time units) much longer than the time interval from which we sampled transients (200 time units). Finally, the model is optimized such that it predicts the right time derivative based on a input variable and its spatial derivatives. So it learns the function mapping from $u, du/dx, d^2u/dx^2 \dots$ to du/dt . This is just the law of a PDE, and it can therefore not memorize any solution/training transients. However, we can expect the model to be valid only close to the smooth attractor. To infer the right space values for new data examples, methods such as Nyström extension can be employed.

Maybe a small point is that we do not learn, or memorize, the dynamics (the behavior as a function of space-time) – we learn the generator, the law that gives rise to the dynamics. That is -if nothing else- a huge compression of the entire space time solution – which now can be regenerated from very little (initial/boundary) information, and even interpolated or extrapolated for nearby conditions.

--Agent based models with complex dynamics abound, much more complicated than those outlined here. For example one area the authors might wish to explore are covid -agent based models (eg VaTech, Northeastern,...). The rules for the agents can be quite complicated, and there is an obvious low order dynamical system at the core. Could the present approach actually be useful in that regard--deriving effective dynamics that are predictive for out of distribution examples?

Yes, we also want to apply our approach to a wider range of agent based models, applying it to the Covid-agent based models mentioned would be an exciting endeavor for us! However, the twist here is (as opposed to earlier works (e.g. Ref [1] below) where a physical space coordinate for an agent-based system is available), we infer here an emergent space in which to learn the PDE. This step is, in a sense, redundant when a space variable is already available! For the present article, we chose a particularly simple model, with the hope to keep the explanation of our approach simple. However, we added now a biologically motivated example as well, to illustrate that our model can be applied to a wider range of systems. For out of distribution examples, if they follow the same governing laws as the training data examples, we are convinced that our approach can be accurate there as well. This, however, excludes cases where some processes for these out-of distribution examples are different (e.g. parameters varied).

[1] Ping Liu, C.I. Siettos, C.W. Gear, I.G. Kevrekidis, Equation-free Model Reduction in Agent-based Computations: Coarse-grained Bifurcation and Variable-free Rare Event Analysis, *Math. Model. Nat. Phenom.* Vol. 10, No. 3, 2015, pp. 71–90 doi: 10.1051/mmnp/201510307

The "dream" behind this work is quite beautiful--but I'm afraid at this point the results could be made much more convincing by choosing better examples.

We added a biologically motivated example of coupled Hodgkin-Huxley type neurons to the article, as mentioned above. This example further illustrates that the proposed approach is indeed invariant to the model details, and that higher-dimensional emergent spaces – and corresponding PDEs – can be learned. We hope you will agree that it goes towards showing the approach is applicable in more complex systems.

REVIEWER COMMENTS

Reviewer #1 (Remarks to the Author):

Many issues of detail that were brought up in my original review have been addressed. But the main issues I raised in the original review still remain.

Because the authors are not learning boundary conditions, they are using simulation data from near the boundaries as the BCs for the learned PDE. But if the data is time-periodic for example, then they are applying a periodic forcing to their PDE and it's no surprise that their PDE gives a periodic solution. Furthermore, the model has no predictive capability since we need data near the boundary to find the solution. So (as with any problem governed by a PDE), for a complete **predictive formulation the boundary conditions are necessary too.**

In my review of the original submission I asked about examples with dynamics more complex than time-periodic. Although the authors added an example in the revision, it also has time-periodic dynamics. In their response, the authors write that "Using chaotic examples would render the comparison between the learned model and the true dynamics even more complicated, because one would have to check that the predicted trajectories diverge with the right Lyapunov exponent and that, at the same time, the attractors of both systems coincide." The authors are correct, but nevertheless other studies exist in the literature that do make comparisons between chaotic data and data-driven models thereof. That could be done here as well.

So as noted originally, there are interesting ideas and methods here. But at this point the formalism is incomplete and most importantly not predictive, and even so has only been applied to problems with simple temporal dynamics. I don't see it as appropriate for Nat. Comm.

Reviewer #2 (Remarks to the Author):

The authors have provided further (interesting) discussion of their methodology and one more application example. I appreciate the extra effort even though the example does not belong into any of the categories that I asked in my first review. Certainly the complexity of the examples can be argued in many different ways. However, I maintain my reservations about the application domains of this methodology. I still find the paper strong in ideas but weak in terms of demonstrating their feasibility to advance the solution of complex problems.

In summary, I recommend publication to Nature Communications on the merit of the ideas presented in the paper. Perhaps applications can follow by these authors or others if this paper gets traction in the community.

Dear Reviewers,

Thank you very much for the feedback on our manuscript "Learning emergent PDEs in a learned emergent space".

We feel we have addressed the point raised by the reviewers by incorporating an example where the dynamics -as requested- is spatio-temporally chaotic (and has periodic boundary conditions, so there is no issue to either learn BC or apply our "narrow corridor" approach). This dynamical state, so called spatio-temporal intermittency, appears in the complex Ginzburg-Landau equation for a suitable set of system parameters. Again, this is not an oscillator or multi-agent example, but we treat the recorded time series obtained from simulations, for illustration purposes, as time series from a discrete ensemble, where the dynamics at each grid point corresponds to a single agent.

As we describe in more detail in the article, we can recover that these [Ⓜ]agents" can be systematically embedded in a one-dimensional periodic emergent domain, and we can successfully learn the effective PDE in this emergent domain. Note that in doing so, we

1. show that we can capture chaotic dynamics with our PDE learning approach, as it can be observed in the new Figure 2,
2. and that, due to the periodic nature of the domain, we do not have to worry about providing and additional information for boundary conditions (diffusion maps in this case showed us that the domain is periodic, see Figure 2). This means that at least here the dynamics are not slaved to any boundary dynamics.
3. Also confirmed that the discovered domain is one-to-one with the original scramble one, thus providing another small validation step for the approach.

We completely agree that the issue of machine learning boundary conditions (in effect, determining what boundary/initial data make a problem well-posed) is an important direction for future research. We are actively working on it – but respectfully, this is an entire research world by itself. Here we learned the space and the equation; we are also working on determining what initial/boundary conditions make the problem for this emergent space/operator pair well posed. We do mention in the Discussion, older initial work by our group (Ref. 43, the "baby-bathwater" scheme) on designing computational experiments to explore how many initial conditions might be necessary).

Again, we are very grateful for the review received, and the ideas and suggestions provided. We hope with the previous inclusion of the network example, and with the current inclusion of a spatio-temporally chaotic example you will consider the paper worthy of publication.

With best regards,

Yannis Kevrekidis

REVIEWERS' COMMENTS

Reviewer #1 (Remarks to the Author):

With the inclusion of the example with chaotic dynamics, I am satisfied that the authors have addressed my comments. I recommend publication.

Reviewer #2 (Remarks to the Author):

I appreciate the discussion added by the authors. I consider the paper acceptable for publication.

REVIEWERS' COMMENTS

Reviewer #1 (Remarks to the Author):

With the inclusion of the example with chaotic dynamics, I am satisfied that the authors have addressed my comments. I recommend publication.

Reviewer #2 (Remarks to the Author):

I appreciate the discussion added by the authors. I consider the paper acceptable for publication.

We appreciate the positive feedback provided by the reviewers, and the constructive feedback that we received during the whole review process.